# Radar near-field sensing using metasurface for biomedical applications
Mohammad Omid Bagheri ✉, Ali Gharamohammadi, Serene Abu-Sardanah, Omar M. Ramahi & George Shaker

Metasurfaces, promising technology exemplified by their precise manipulation of incident wave properties and exquisite control over electromagnetic field propagation, offer unparalleled benefits when integrated into radar systems, providing higher resolution and increased sensitivity. Here, we introduce a metasurface-enhanced millimeter-wave radar system for advanced near-field bio-sensing, underscoring its adaptability to the skin-device interface, and heightened diagnostic precision in non-invasive healthcare monitoring. The low-profile planar metasurface, featuring a phase-synthesized array for near-field impedance matching, integrates with radar antennas to concentrate absorbed power density within the skin medium while simultaneously improving the received power level, thereby enhancing sensor signal-to-noise ratio. Measurement verification employs a phantom with material properties resembling human skin within the radar frequency range of 58 to 63 GHz. Results demonstrate a notable increase of over 11 dB in near-field Poynting power density within the phantom model, while radar signal processing analysis indicates a commensurate improvement in signal-to-noise ratio, thus facilitating enhanced sensing in biomedical applications.

Radar systems have recently been used in many bio-sensing applications to extract specific bio-signals pertaining to the target individual's health. They can be successfully applied to contactlessly characterize a number of biomedical parameters, detect emergencies, and provide excellent long-term care benefits[1]. Over the last decade, several prototypes have been developed that demonstrate the potential of using AI-powered radar systems for non-invasive glucose sensing[2], wearable sweat monitoring[3,4], multi-person vital sign tracking[5–7], gait monitoring[8], fall detection[9], human eye activity monitoring[10], and imaging[11]. Compact radars are also often used in wearable technology, such as smartwatches, for routine human health monitoring[12,13]. Radar system development for industrial and biomedical microwave applications has recently accelerated due to an increase in interest in millimetre (mm)-wave communication for near-field sensing using low-profile and low-cost antennas providing sufficient energy penetration into the body[14]. Recently published papers in peer-reviewed journals demonstrate the utility of some compact mm-wave off-the-shelf short-range radar modules in biomedical applications such as blood glucose concentration level detection[15], differentiating blood samples from disparate glucose concentrations[16], breast cancer detection[17], skin cancer detection[18], arterial pulse waveforms measurements for blood pressure tracking[19], and continuous extraction of cardiorespiratory displacement waveforms with high precision[20].

Characterized by elevated blood glucose levels, diabetes is a prevalent and chronic condition that underscores the crucial need for early detection and diagnosis. Despite the availability of invasive techniques, the increasing focus on non-invasive glucose measurement, offering additional benefits without causing inconvenience to the human body, drives ongoing research and the exploration of new possibilities[21]. Nonetheless, monitoring blood glucose non-invasively remains challenging, with no reliable commercial or clinical device developed from various explored techniques[22]. A portable planar microwave sensor, operating within the Industrial, Scientific, and Medical (ISM) band at 2.4–2.5 GHz, is introduced to facilitate non-invasive monitoring of blood glucose levels[2]. In an alternative approach to glucose monitoring, a robust, low-power millimeter-wave radar system is employed to detect varying glucose concentration levels in artificial blood samples, thereby enabling differentiation[15,16]. Another method involves utilizing an antenna sensor operating at 4.2 GHz, positioned on the pancreas to capture dielectric radiation signals associated with glucose levels[22]. Additionally, a microwave biosensor for real-time non-invasive glucose monitoring operating in the range of 1–6 GHz is developed[23]. These recent advancements collectively underscore the increasing demand for the development of a high-sensitivity radar system to enable continuous monitoring of blood glucose levels.

When considering the hardware design of a radar system, the radar chipset architecture and an appropriate design of the accompanying

Department of Electrical and Computer Engineering, University of Waterloo, Waterloo, ON, Canada. ✉e-mail: omid.bagheri@uwaterloo.ca

antenna are required. When used in bio-sensing applications, the performance of a radar hardware system is evaluated by utilizing the antenna's near-field sensing capabilities when placed in close proximity to a human body. Near-field-focused (NFF) antenna design has been studied in the literature for a long time for body-centric wireless communications applications[24,25]. Utilizing the radiated power of the transmitter antenna by focusing the electric field at a specific location close to the radar surface is an effective method for improving near-field sensing[26].

Different types of antennas, such as high-profile reflectors or aperture antennas are frequently used to provide high sensing in the near-field region[27]. The quadratic phase for near-field focusing on reflector antennas can be obtained by defocusing the feed away from the focal point[27]. The structure produced by this method, however, is bulky and unsuitable for low-profile and compact applications. Planar antennas with a low profile and easy-to-fabricate features present an alternative option[28]. Two facing microstrip patch antennas, transmitter (TX) and receiver (RX), operating at 60 GHz, are employed in describing a near-field sensing system for glucose level monitoring[29]. However, the single microstrip planar antenna essentially provides low field intensity in the near-field region. To address this challenge, a near-field focused microstrip array antenna operating at 2.45 GHz is proposed for maximizing the power transmission efficiency between two antennas[30]. However, a large number of array elements are necessary to achieve high focused power in the near-field region resulting in a high-profile structure with a complex feeding network design, high fabrication cost and loss.

Space-fed planar array antennas, such as reflectarrays and transmitarrays, offer an alternative by removing the intricate and expensive feeding networks, resulting in higher efficiency by modifying the phase front of the radiated fields from the feed[31]. Examples illustrating this concept include an optically transparent reflectarray antenna operating at 5.8 GHz[32] and a bi-layer resonant wires transmitarray antenna at 157 MHz[33], both designed to enhance wireless power transfer in the near-field. Moreover, near-field multi-focus reflectarray and transmitarray apertures, illuminated by numerous horn antennas, are designed with a distance of 9.6 $\lambda$ at 5.8 GHz[34] and 9.3 $\lambda$ at 28 GHz[35]. A folded transmitarray structure fed by a horn antenna at a spacing of 2.2 $\lambda$ is proposed to concentrate power in both the near- and far-field at 2.45 GHz[36]. Since the feed is not in the near-field radiation zone, the feeding blockage has no impact on the radiation of the transmitarrays in comparison to the reflectarrays[37]. In the literature on near-field focusing, existing transmitarray antenna designs do not possess the capacity to seamlessly integrate with a low-profile radar system operating at mm-wave frequencies. This limitation is attributed to their conspicuous high-profile structures, along with their reliance on a bulky feed antenna positioned at a considerable distance from the aperture.

To get around the current limitations of bioelectronic interfaces, metasurfaces with subwavelength structures can be engineered to control electromagnetic fields around the human body[38]. Integrated metamaterials[39] and superstrate structures with reflective or transmissive functionality are utilized in near-field sensing by invoking the arrays' mechanism and taking advantage of the compact feeding antennas' design compatibility to offer suitable features in the near-field region[26]. Particularly, an engineered superstrate with two layers of perforated aluminium sheets is used as a near-field manipulator to adjust the electromagnetic radiation pattern of a horn antenna by modifying its near-field components in the frequency range of 10 GHz to 12 GHz[40]. In addition, a transmissive programmable metasurface for high-resolution far-field and near-field detection is reported in the literature[41]. Each metasurface unitcell is made up of five microstrip layers with an air gap in between, and a programmable phase shifter is used to adjust the phase shift for various focusing points at 5.75 GHz. These approaches are not suitable for mm-wave frequency bands, and the proposed structures have a complex design, high profile, and high manufacturing costs. Based on our extensive review of the literature, we have observed that none of the metasurfaces documented so far possesses a suitable structure that combines low-profile and high-integration capability, particularly to accommodate mm-wave frequencies. This

deficiency hinders their integration with radar sensors, thereby limiting their potential for enabling high-precision sensing in near-field biomedical applications.

Additionally, the majority of studies on near-field focusing consider the antenna's radiation in free space. The human body creates a different environment in the antenna's radiation region, which leads to detuning and impedance mismatching, which degrades the antenna performance and lowers the penetration power density into the body, making the prior methods unsuitable for biomedical applications when the human body is in contact with the radiating elements[13]. To address this issue, some efforts in literature have been reported. The method involves treating the skin as a layer of the antenna substrate operating at 2.45 GHz, aiming to decrease the signal scattering from the skin and direct more of the transmitted signal towards the tumor[42]. In a recent study, this issue is addressed by introducing a chest-wearable 60 GHz radar system for continuous monitoring of cardiorespiratory displacement waveforms, evaluating the antenna's performance in close proximity to the skin with an air gap of $\lambda$ using electromagnetic simulations[20]. Placing an appropriate solid or liquid media between the antenna and the tissue is another approach to effectively reduce antenna mismatching[43]. To mitigate impedance mismatch, however, a number of media layers are required, resulting in a high-profile structure that poses integration difficulties, and applying liquid onto a person's skin can evoke discomfort. Metamaterials and metasurfaces, which are more practical and easier to integrate with radars, can also be used to improve antenna matching in the presence of tissue[44–47]. A bow-tie antenna design utilizing metasurfaces as an impedance-matching layer between the biological tissue and the antenna is proposed to enhance radiating performance when close to the human body at 2.56 GHz[13]. However, creating a metasurface with the intention of enhancing antenna matching close to the human body is insufficient to deliver a highly near-field-focused power directed towards the body, and to significantly enhance the near-field sensing capabilities of the radar.

The concept of near-field sensing improvement of a radar system including transmitter/receiver (TX/RX) antennas implies a significant increase in the near-field power density radiated by TX into the body without disturbing antenna impedance matching while improving radar sensing by providing more received power reflected from the body into the RX antenna, representing the signal-to-noise ratio (SNR) enhancement of the radar system. It is therefore extremely challenging to develop a compact, wireless mm-wave radar for biomedical applications meeting the proposed requirements for high near-field sensing, high levels of integration, and low manufacturing costs.

Given a specific mm-wave radar chipset and antenna design, along with a pre-determined human body area of interest, Fig. 1a, this paper presents an approach for the high near-field sensing of the radar system for biomedical applications by integrating a designed planar transmissive metasurface acting as a buffer between the radar TX/RX on-chip antennas and the human body skin as depicted in Fig. 1b. The presented methodology, metasurface-enhanced radar near-field sensing, could be applied to a diverse set of biomedical sensing applications, including glucose monitoring[2,15,16], skin cancer detection[18], as well as heart and on-body radar cardiorespiratory monitoring[20]. This paper highlights its focus on real-time disease diagnostics, particularly emphasizing the crucial role of continuous blood glucose monitoring for diabetes diagnosis, aiming to integrate glucose monitoring into wearable devices, showcasing the specialized use of this metasurface technology. Full-wave electromagnetic simulator is used to assess the near-field performance of the radar antenna integrated with and without the metasurface in direct contact with a human skin model. Antenna impedance matching, signal-to-noise ratio, and electromagnetic field penetration into the human skin are performance parameters that need to be quantified.

## Methods
### Near-field-focused metasurface design in skin medium
In this section, we will discuss the design principles of the proposed planar transmissive metasurface, which is aimed at concentrating the near-field

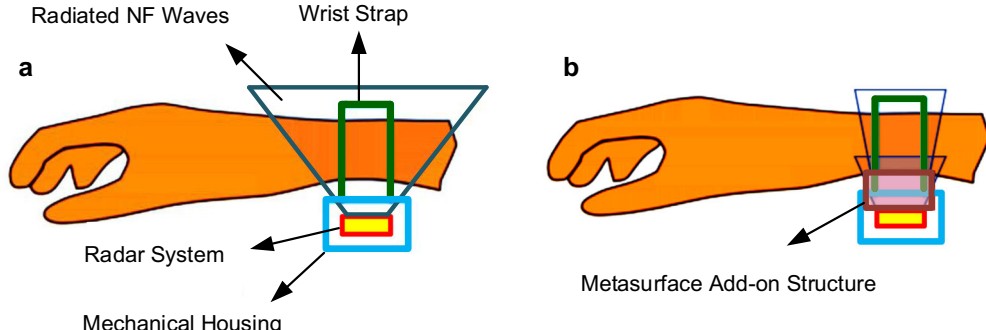

**Fig. 1 | Enhancing wearable radar sensing with metasurface technology near the human hand. a** A wearable device equipped with radar technology positioned near a human hand. **b** Incorporating a metasurface structure between the radar antennas and the human hand model to enhance the near-field (NF) energy coupling and signal-to-noise ratio (SNR).

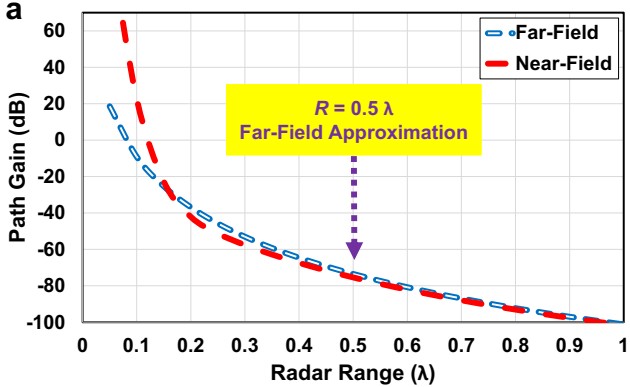
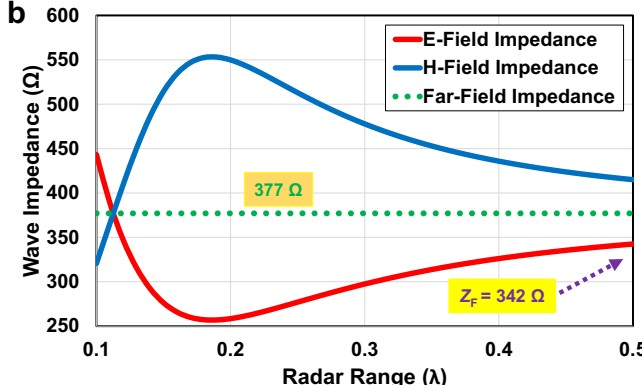

**Fig. 2 | Comparative analysis of path gain and wave impedance in near-field and far-field regions of radar antenna. a** Investigation of the path gain versus radar range variations for the near-field in comparison to the far-field. **b** Variations of the free-space wave impedance in the near-field region of the radar antenna, depicted in terms of the electric (E) and magnetic (H) field.

power emitted from the radar transmitting antenna into a human skin model, while simultaneously enhancing the power captured by the radar receiving antenna, thereby improving the overall system sensing capabilities.

The design process and performance of the metasurface are examined, taking into account the design of the unitcell and the analysis of the phase-synthesized array. The fundamental concept, as per transmitarray theory, involves adjusting the phase of the electric field radiation or the current sources of the array elements on the antenna's aperture. This adjustment is done in such a way that their contributions converge in phase at a specific focal point in the near-field region. In the context of near-field-focused antennas, the radiated field's maximum is achieved near the aforementioned focal point. This implies that the peak of the power density is positioned between the focal point and the antenna's aperture[14]. This is accomplished by using symmetric source-phase tapering, which compensates for the various distances between each source point on the aperture and the focal point. On the other hand, to increase the received power reflected from the human skin model, impedance matching analysis in the metasurface design is highly required. The following discussion covers the theory for the radar antenna radiation in the near-field region and unitcell and phase-corrected array designs in the presence of the human skin model.

**Near-field radar radiation**

To investigate the radar performance based on the theory in the scenarios with and without the metasurface, one can use the Friis radar equation applicable to the near-field region by modifying the parameters for the near-field analysis, including the antenna gain and radar range equation[48]. For traditional monostatic radar systems with far-field radiation, the radar path gain is given as,

$$\text{Path Gain} = \frac{P_{RX}}{P_{TX}} = \underbrace{\frac{G_{TX}}{4\pi R^2}}_{TX} \underbrace{\frac{G_{RX}\sigma}{4(kR)^2}}_{RX} \tag{1}$$

Where $P_{TX}$ and $P_{RX}$ are the transmitted and received power, $G_{TX}$ and $G_{RX}$ are the transmitter and receiver antenna gain, respectively, $k$ is the wave number, $\sigma$ is the radar cross section and $R$ is the radar range between the TX and RX antennas.

According to (1), Friis's Law states that as the range increases, the transmitted and reflected power density in the far-field diminishes proportionally to the inverse square of the distance as expressed in (2). However, when it comes to the near-field region, electric and magnetic fields behave differently such that this principle needs to be modified as the power decreases at a faster rate than the inverse square[49]. Therefore, the available power in a near-field link is much higher than the usual far-field. Modifying (1) using the near-field consideration related to the radar range shows that the received power is proportional to the near electric field as (3). Figure 2a provides a comparison of path gain variations relative to radar range, contrasting the far-field and near-field scenarios as represented by (2) and (3), respectively. It demonstrates that at very short ranges, the difference between path gains is in the order of +40 dB approximately. After the range of 0.2 λ, the path gain variations versus the radar range for the near-field, however, is nearly identical to that of the far-field.

$$P_{RX} \sim |E|^2 \sim \left[\frac{1}{(kR)^2}\right] \text{Far-Field} \tag{2}$$

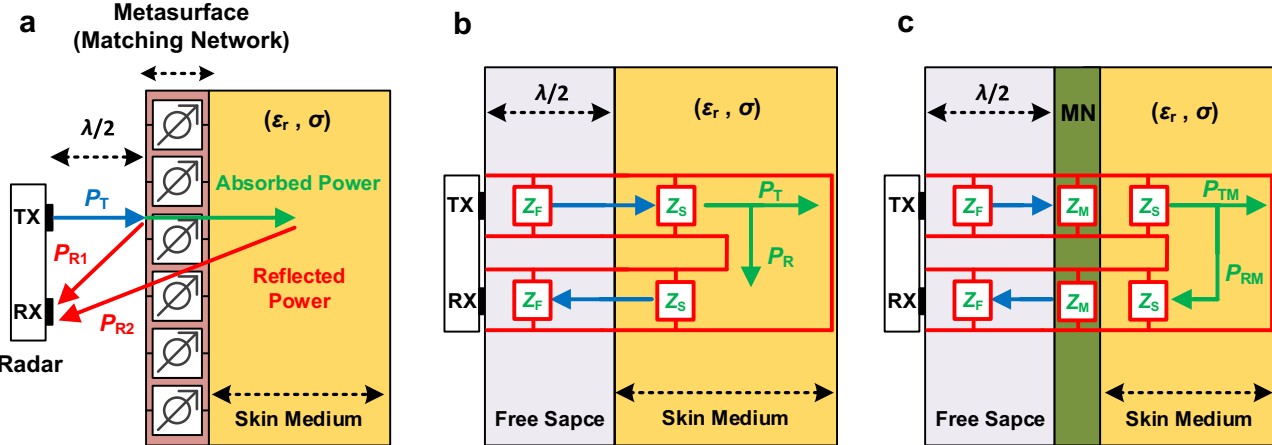

**Fig. 3 | The transmission line model analysis for the radar system in the presence of the human skin model and metasurface integration. a** Analysis of transferred and received power between transmitter (TX) and receiver (RX) antennas in the metasurface-enhanced radar system loaded by skin model. **b** Investigation of the radar system performance using impedance analysis for each medium, comparing the radar system without a metasurface to (**c**) the radar system with integrated metasurface.

$$P_{RX} \sim |E|^2 \sim \left[ \frac{1}{(kR)^2} - \frac{1}{(kR)^4} + \frac{1}{(kR)^6} \right] \text{Near - Field} \quad (3)$$

$$G_{TX}, G_{RX} = \frac{\text{Re}(\tilde{E} \times \tilde{H}^*) \cdot 2\pi R^2}{\text{Input Power}} \quad (4)$$

To achieve a concentrated near-field power, the radar range which is the distance between the feeding antenna and the metasurface in contact with the skin model is crucial. If an air gap distance of $\lambda/2$ ($\lambda$ in free-space) is considered, in accordance with the Fabry-Perot Cavity (FPC) theory, the proposed metasurface is employed in the configuration of a FPC which reduces backscattering and increases the peak radiation of the underlying source antenna leading to enhance transferred power density into the skin[50]. As a result, placing the metasurface and skin model at the range of $\lambda/2$ at 60 GHz, 2.5 mm air gap above the radar antenna, allows using far-field approximation, and utilizing Friis's Law with the transmitted and reflected power variations proportionally to the inverse square of the range, (2), is acceptable.

In the near-field region, where the electromagnetic wavefronts are not far enough apart to be considered planar waves, the gain of an antenna can be defined in terms of the ratio of the power density in a particular direction to the power density that would be produced by an isotropic radiator at the same distance and with the same input power[51]. Considering the above-mentioned far-field approximation, the relationship between the near-field axial field strength and the near-field gain can be expressed as (4). The axial field strength of the antenna in the near-field can also be obtained through the full-wave simulation. Thus, increasing the electric field intensity in the near-field region of the antenna using the transmissive metasurface, has a direct impact on the near-field gain improvement at the specific distance and particular direction for both TX and RX radar antennas. As a result, according to (1), the near-field gain improvement leads to the received power, $P_{RX}$, enhancement which provides better sensing for the proposed mm-wave radar.

**Metasurface as a matching medium**

In accordance with the transmitarray theory, using a transmissive meta-surface between the radar system and the skin model can provide higher focused transmitted as well as received power to increase SNR. In such instances, the functional role of the metasurface as an impedance matching interface between the radar antenna radiation and the human skin medium is of paramount significance, particularly within the domain of close-range radar applications featuring proximate positioning of metasurfaces along-side radar antennas. In this context, analysis of the metasurface's spatial placement is imperative to determine whether it resides within the far- or near-field region of the radar antenna's radiation.

In the far-field region, the electric and magnetic waves move together with synchronized phases and amplitudes fixed by the impedance of free space, $120\pi \, \Omega$. However, in the near-field region, as indicated in Fig. 2b, these fields are out of phase and the ratio of electric to magnetic field amplitudes is a strong function of both radial distance to the source and orientation, resulting in non-uniform wave impedance deviation from the standard characteristic impedance of $377 \, \Omega$.

When integrating a metasurface into the radar system, the radar transmitter (TX) antenna radiates power $P_T$ into the near-field region directed towards the metasurface loaded with a human skin model. Con-sequently, the power received by the radar RX antenna comprises reflections from the skin medium and the metasurface surface, $P_{RX} = P_{R1} + P_{R2}$, as shown in Fig. 3a. In this case, the impedance matching considerations are required in the design of the metasurface such that the reflected power from the metasurface, $P_{R1}$, is decreased and $P_{RX}$ mostly provides the information coming from the skin medium leading to higher biomedical sensing. To this end, the impedance of the skin medium and the propagated wave in the near-field region of the free space are required.

In such a case, the general wave impedance equations can be used for the near-field region as expressed in (5)[52]. Where $\eta_0$ is the free-space impedance in the far-field region, $\beta$ is the propagation constant and $R$ is the distance from the radar antenna surface.

$$Z_E(r) = \eta_0 \times \frac{|1 + \frac{1}{j\beta R} + \frac{1}{(j\beta R)^2}|}{|1 + \frac{1}{j\beta R}|} \quad , \quad Z_H(r) = \eta_0 \times \frac{|1 + \frac{1}{j\beta R}|}{|1 + \frac{1}{j\beta R} + \frac{1}{(j\beta R)^2}|} \quad (5)$$

$$Z_L = \sqrt{\frac{\mu_0}{\varepsilon_0 \varepsilon_r (1 - \frac{j\sigma}{\omega \varepsilon_0})}} \quad (6)$$

As investigated, a half-wavelength distance considered in this paper, makes the electric and magnetic field phases diverge and as shown in Fig. 2b, the wave impedance in near-field free-space, $Z_E(r = \lambda/2) = Z_F$, is obtained as $342 \, \Omega$ at the proposed distance which is close to the free-space wave impedance in the far-field region, $377 \, \Omega$. In addition, as expressed in (6), $Z_L = Z_S$ is defined as the impedance of a general lossy dielectric medium which can be used as the biological load impedance for the skin model[13].

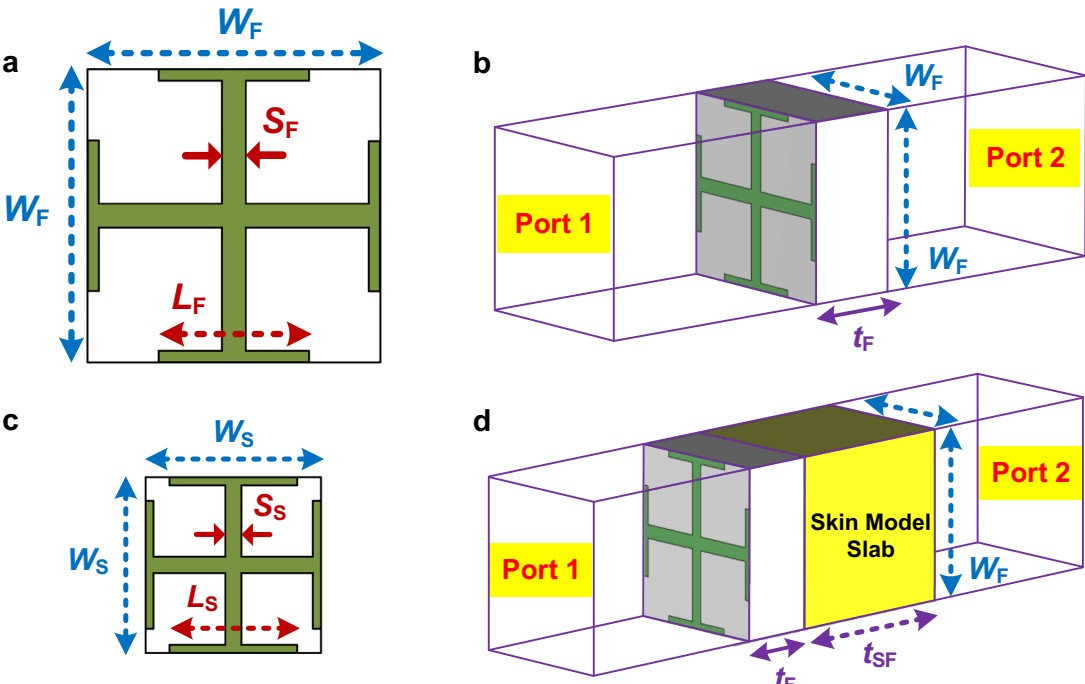

**Fig. 4 | Comparative analysis of transmissive metasurface unitcells designed in free-space and human body skin medium. a** Unitcell dimensions designed for radiation in free space. **b** Simulation representing the Floquet port analysis of the unitcell designed for free-space. **c** Unitcell designed for radiation in human body skin medium. **d** Simulation illustrating the Floquet port analysis of the unitcell in the presence of the human body skin modelled as a dielectric slab. The design parameter values are: $W_F = 2.5$, $t_F = 1.25$, $L_F = 1.32$, $S_F = 0.2$, $W_S = 1.1$, $L_S = 0.65$, $S_S = 0.2$, $t_{SF} = 3$ (all in mm).

Given $Z_F$ and $Z_S$, the transmission line model is a suitable option to consider impedance matching analysis in the metasurface design[13]. To achieve this, the human body skin situated in close proximity to the radar antenna is simplified as a homogenous dielectric slab possessing the characteristic properties of the analyzed anatomical skin region at 60 GHz. The impedance of this slab is determined through the utilization of (6). The transmission line using the equivalent circuit model for the radar system integrated with and without metasurface is presented in Fig. 3b, c. In Fig. 3, $P_T$ and $P_R$ denote the power that is transmitted through and reflected from the skin medium in the absence of the metasurface. Conversely, $P_{TM}$ and $P_{RM}$ symbolize analogous power quantities in the presence of the metasurface matching network. Furthermore, $Z_F$ is the free-space wave impedance in the near-field region; however, at the point of metasurface interference, $\lambda/2$ distance from the antenna surface, the exact value of impedance can be used, or it can be approximated with the free-space wave impedance in the far-field as explained above. In that case, the impedance of the metasurface can be adjusted as $Z_M$ by simulating an impinging plane wave upon the proposed transmissive unitcell.

**Planar metasurface unitcell design in skin medium**
It is necessary to conduct a design analysis of a periodic unitcell as a transmitting element in order to streamline the design cycle of the proposed transmissive metasurface. The square unitcell used in this study consists of two metallic layers printed on each side of Rogers RO4003, a dielectric substrate, with 1.25 mm thickness. As seen in Fig. 4a, the metallic layer is created by a symmetric crossed planar electrical dipole that offers dual-linear polarization, and each branch size can be altered to control the transmitted fields' phases. A full-wave simulator, HFSS, is used to evaluate the unitcell impedance matching at 60 GHz as shown in Fig. 4b using the Floquet port analysis for the infinite array design.

In the realm of biomedical applications, achieving an optimal design to enhance the penetration of electric field intensity into the human body skin necessitates meticulous consideration of the skin's actual model during unitcell design and related simulations, to ensure proper impedance matching when the unitcell interfaces with skin media, rather than free space. Accordingly, the human body skin is modeled as a dielectric slab with a thickness of 3 mm, incorporating the characteristics of human skin which typically exhibits a permittivity of 7.98 and a conductivity of 36.4 S/m at 60 GHz[53], to be contacted with the metasurface unitcell without any air gaps in between.

According to the microstrip design theory, adding a dielectric slab with $\epsilon_r = 7.98$ instead of $\epsilon_r = 1$ which is used for free-space medium, increases the effective permittivity of the total environment roughly from 2.2 for the unitcell with free-space, to 5.7 for the unitcell with body skin model providing the advantage of the smaller unitcell dimensions leading to low-profile array while reducing phase error losses. In the design of the unitcell working in free-space, Fig. 4a, the typical values of the length and width are considered as $0.5\lambda$ ($\lambda$ is the free-space wavelength at 60 GHz) with a 1.25 mm substrate thickness, while the dimensions of the designed unitcell with human body skin model reduce to $0.22\lambda$ with a 0.8 mm substrate thickness, Fig. 4c. The other advantage of unicell size reduction is to ensure that $W_F < \lambda$ avoiding more than one spot region appearing in the near-focused region, as a similar concept to the grating lobes occurs in the far-field radiation pattern of any uniformly spaced array[14]. Figure 4d illustrates a simulation model representing the Floquet port analysis of both unitcells in direct contact with the human body skin, modeled as a dielectric slab.

Figure 5a illustrates the reflection coefficient analysis of the designed unitcell operating in free space compared to the human skin medium, as depicted in Fig. 4b, d respectively. The analysis in Fig. 5a highlights a significant challenge encountered in the unitcell design, where reflection from the air-skin interface leads to notable mismatching, distinguishing it from typical metasurface designs reported in the literature. In this study, the modified version of the designed unitcell specifically tailored for in-body skin radiation, as shown in Fig. 4c, effectively addresses this issue.

From the reflection coefficient analysis, the unitcell impedance variation in the frequency range of interest is extracted and shown in Fig. 5b. Comparing wave impedances inside the human skin model, $Z_S$, and wave impedance in the metasurface, $Z_M$, indicates that the design parameters of

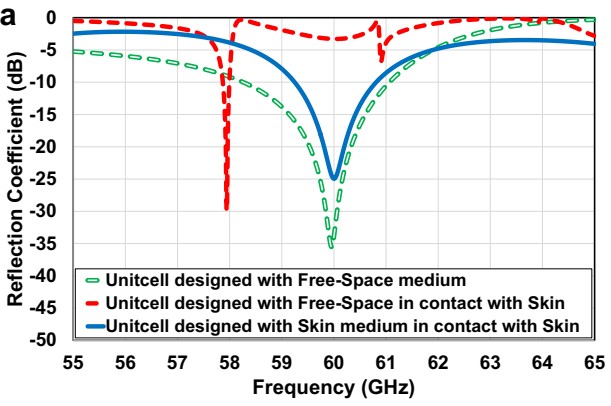

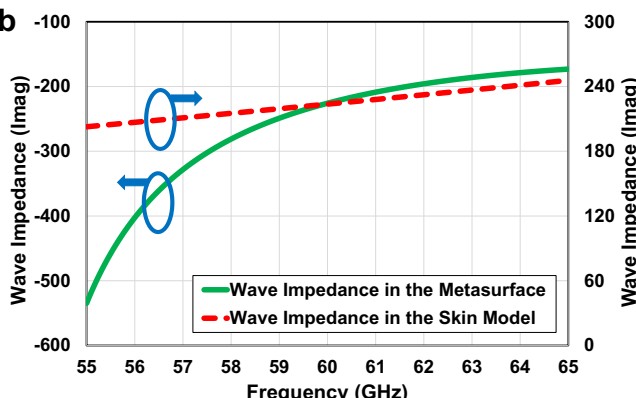

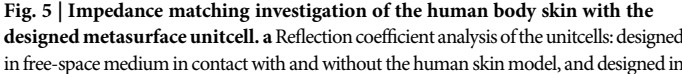

**Fig. 5 | Impedance matching investigation of the human body skin with the designed metasurface unitcell. a** Reflection coefficient analysis of the unitcells: designed in free-space medium in contact with and without the human skin model, and designed in the skin medium in contact with the human skin model. **b** Imaginary part variations of the wave impedances versus frequency for the skin model and the designed metasurface.

the metasurface are optimized such that providing a capacitive behavior within the desired frequency band and, in particular, a value close to the skin model impedance but opposite in sign at 60 GHz. Therefore, the designed metasurface unitcell is able to counteract the inductive behavior of the human skin model providing high impedance matching at this frequency range as shown in Fig. 5a. As a result, using the designed transmissive metasurface with skin features consideration reveals a high impedance matching at 60 GHz between the radar antenna propagation and the skin model leading to a higher transferred power into the skin medium, $P_{TM} > P_T$, and higher reflected power received by RX antenna, $P_{RM} > P_R$.

## Planar metasurface array design
The metasurface layer is a structure where a wave from a feed is incident on a planar array of elements. In order to produce the desired beam shape on the opposite side of the planar array in a particular location in the near- or far-field region, each element is created to add a particular phase shift to the incident electric field while maintaining magnitude in a level close to the maximum. Considering a square planar aperture consists of $N \times N$ radiating elements, with an inter-element distance $W_S$ along both axes, the (m, n)th element indicates the center point location of each unitcell in the x-y plane, where $m$ and $n$ are integers, varying between $(-N+1)/2$ and $(N-1)/2$, $N$ is considered to be an odd number. The electric field radiated at an observation point $P(x, y, z)$ by the $N \times N$ array expressed as[14],

$$E(p) = \sum_{m=(-N+1)/2}^{(N-1)/2} \sum_{n=(-N+1)/2}^{(N-1)/2} I_{mn} E_{mn}(p) \tag{7}$$

Where $I_{mn}$ and $E_{mn}(p)$ are the excitation coefficients and the radiated electric field corresponding to the $(m,n)$th element of the array, respectively. For each element, the feeding-current amplitude is adjusted equal to $I_0$, while its phase, $\phi_{mn}$, is selected so that all the element contributions add in phase at the focal point $F = (0,0,r_0)$, located along the normal to the array plane,

$$\phi_{mn} = \frac{2\pi}{\lambda}\left[\sqrt{(x_m^2 + y_n^2 + r_0^2)} - r_0\right] + \phi_0 \tag{8}$$

Where $\phi_{mn} = \phi_0$ provides the uniform-phase unitcells in planar array making far-field-focused radiation. When the focal distance, $r_0$, is greater than the aperture size, the above phase tapering is usually approximated with a quadratic law which is valid for the far-field region[54]. When the observation point is close to the array elements, $2\pi R/\lambda < 1$, where $R$ is the distance between the radiating element and the observation point, the far-field considerations such as parallel-ray approximation are no longer valid

to use in the electric field equation presented in (7). Additionally, the exact phase shift calculation as described in (8) is necessary to focus the electric field at the focal point $r_0$, which is situated in the near-field region. Approximation formulas result in phase error losses and non-focusing in the near-field.

The proper dimensions of the array unitcells are obtained by finding the required phase delay compensation of the electric field at the frequency of interest using (8) in the planar aperture at the metasurface location. This phase compensation mechanism results in an equal-phase superposition of the propagated fields at the focal point in the near-field region.

Using (8), the required E-field phase compensation ranges on the aperture supposed in the metasurface location at $r_0 = 2.5$ mm are obtained and the accessible phase ranges in x and y directions according to the number of the array elements, from $3 \times 3$ to $9 \times 9$, are investigated and shown in Fig. 6. The required phase shift for each array, illuminated by a normally incident wave radiating from a feed aligned with the center of the array, is obtained as $\Delta\phi(3 \times 3) = 30$, $\Delta\phi(5 \times 5) = 120$, $\Delta\phi(7 \times 7) = 210$, $\Delta\phi(9 \times 9) = 340$ degrees which are presented in Fig. 6a to d, respectively. Figure 6e, f depict a comparative analysis of the required phase shift compensation when the feed is offset from the center of the array compared to when the feed is positioned at the center. While using more unitcell elements in the metasurface structure increases the electric field intensity, choosing the number of array elements is constrained by the acceptable dimension as determined by the feeding antenna and, above that, the range of the phase range compensation that unitcell can offer.

To design a full planar array metasurface, unitcells that can deliver 360 degrees of phase shift are required to cover the required phase shift range with a minimum at the center element, $\phi_{mn} = \phi_0$, and maximum at the edge elements[55]. The maximum phase shift that a single-layer transmitarray metasurface unitcell can provide is 90° for − 3 dB transmission coefficient regardless of the shape of the conducting element[56]. To increase the amount of phase shift, stacked multiple single-layer unitcells can be used. for the −3 dB transmission bandwidth, the two-layer structure provides a phase shift of up to 180°, whereas the three-layer structure provides a phase shift of up to 300°. Increasing the number of layers also increases the bandwidth for the desired frequency band[57], while increasing design complexity and manufacturing cost.

According to this, the designed unitcell in this paper is expected to provide up to 180° phase shift at the magnitude reduction of up to −3 dB due to using two metallic layer structures. By changing the length of the metallic crossed dipoles' branches, $D_S = L_S + W_S/2$, it is demonstrated that the proposed unitcell can provide the desired phase shift range at 60 GHz as shown in Fig. 7a. Considering Fig. 6, based on the 180° phase shift range that the unitcell provides, a suitable number of the array elements should be selected by including the largest possible aperture to cover the source

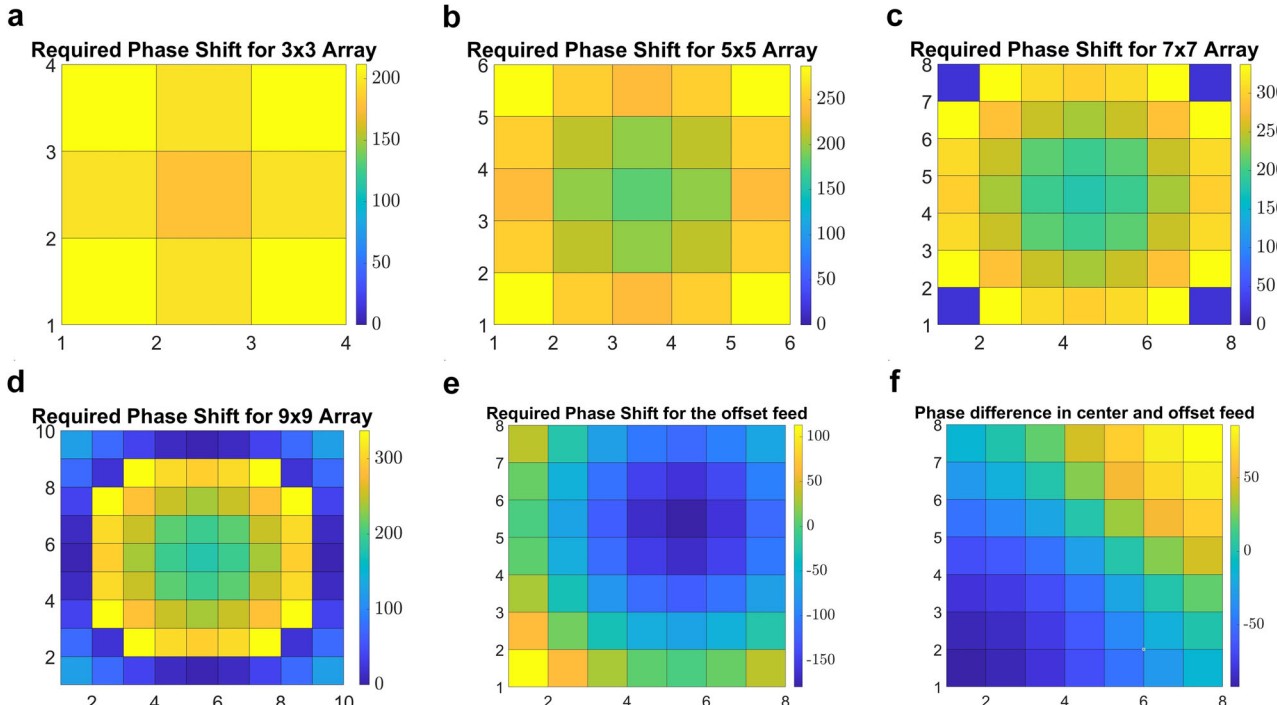

**Fig. 6 | Investigation of unitcell required phase shift to design a near-field-focused transmitarray metasurface to be placed at $\lambda/2$ distance from the feed based on the number of the elements in x and y directions. a** $3 \times 3$ elements (**b**) $5 \times 5$ elements (**c**) $7 \times 7$ elements (**d**) $9 \times 9$ elements, (**e**) the required phase shift compensation of the $7 \times 7$ array when TX radar source is in the offset, and (**f**) the phase difference between the case when source is placed in offset with the case source is placed at the center of the $7 \times 7$ array.

antenna properly and provide the highest accessible focused power without substantially increasing the phase error loss. Figure 6a, b show that the $3 \times 3$ and $5 \times 5$ array elements with required phase compensation of 30° and 120° are not quite effective in a focusing process. Moreover, the $9 \times 9$ array with required $\Delta\phi = 340°$ shown in Fig. 6d, makes a large phase error loss due to unitcell phase limitations. Among all options investigated, the array of $7 \times 7$ with $\Delta\phi = 210°$ is the closest one to the available 180° phase shift as shown in Fig. 6c. Therefore, the $7 \times 7$ array utilizing 49 phase-corrected elements provides the highest power density into the human skin.

The variations of the unitcell transmission coefficients, phase and magnitude, versus incident wave angle, are also investigated and shown in Fig. 7b. As shown, taking advantage of the low-profile unitcell with $0.22\lambda$ dimension, when the unitcells are illuminated at various locations, the oblique incidents have a very low impact on the magnitude and phase of the unitcell. From Fig. 7b, for the proposed metasurface placed at $\lambda/2$ distance, the maximum incidence angles up to 70° required at the edge elements causes the phase deviation lower than 4°, while the magnitude makes only 0.04 dB variations. Therefore, the phase error loss caused by the oblique incident is negligible, which simplifies the design process while guaranteeing nondestructive effects on the phase difference compensation.

The preceding investigation pertains to the metasurface unitcell and array, with the analysis conducted at the central frequency of 60 GHz. In light of the designated bandwidth from the source antenna, the proposed radar chipset spanning 58 to 63 GHz, it becomes imperative to assess the fluctuations in transmission coefficients across the entire frequency spectrum within the specified range. This comprehensive evaluation is essential for ascertaining the overall bandwidth of the entire system. Figure 7 illustrates the magnitude and phase responses of the two-layers' unitcell for varying lengths of crossed dipole metallic branches. The results are presented across different frequencies within the bandwidth of interest. In Fig. 7c, it is observed that the frequency range from 59 GHz to 61 GHz exhibits a favorable transmission magnitude change, with a maximum reduction of 1 dB. Conversely, at 62 GHz, a higher reduction exceeding 3 dB is observed, indicating that the power improvement technique is less

effective at this frequency. However, merely examining the transmission coefficient's magnitude is insufficient for determining the bandwidth of the metasurface-enhanced radar design. Consequently, the investigation extends to the transmission coefficient's phase, as depicted in Fig. 7d within the desired bandwidth. Notably, the phase response for frequencies ranging from 59.5 to 61.5 GHz appears nearly parallel, providing a constant shift throughout the bandwidth. Given the satisfactory results obtained from both magnitude and phase analyses of the transmission coefficient in the frequency range of 59.5 to 61.5 GHz, it is anticipated that the near-field-focused radar antenna integrated with the designed metasurface, affords a 2 GHz bandwidth to enhance near-field power within the skin.

## Results and discussion
### Near-field analysis of the radar system integrated with and without metasurface
Due to the demand for a compact and low-profile radar structure in biomedical applications, a suitable choice for near-field sensing improvement is using the planar array antennas consisting of a source and a transmissive surface. According to transmitarray theory, considering a microstrip PCB antenna as a source, one can improve its near-field performance in the presence of the human body by adding a surface acting as an impedance-matching network between the source and the human skin[25]. The proposed theory is applicable to any type of radar system utilizing planar TX/RX antennas-on-chip.

As shown in Fig. 8a, the Infineon BGT60TR13C radar chipset includes four on-chip antennas, one of which radiates as a transmitter and the others of which serve as receivers[58]. The radar TX antenna can be considered as a feed of the transmitarray, and the planar metasurface located above the radar is used as a transmitting surface. In this section, the designed planar transmissive metasurface is used and it is investigated how the Infineon BGT60TR13C radar chip antenna interacts with and without the proposed metasurface regarding free-space and skin impedance matching, near-field-focused power absorbed from the transmitter antenna into the human skin, and radar SNR improvement.

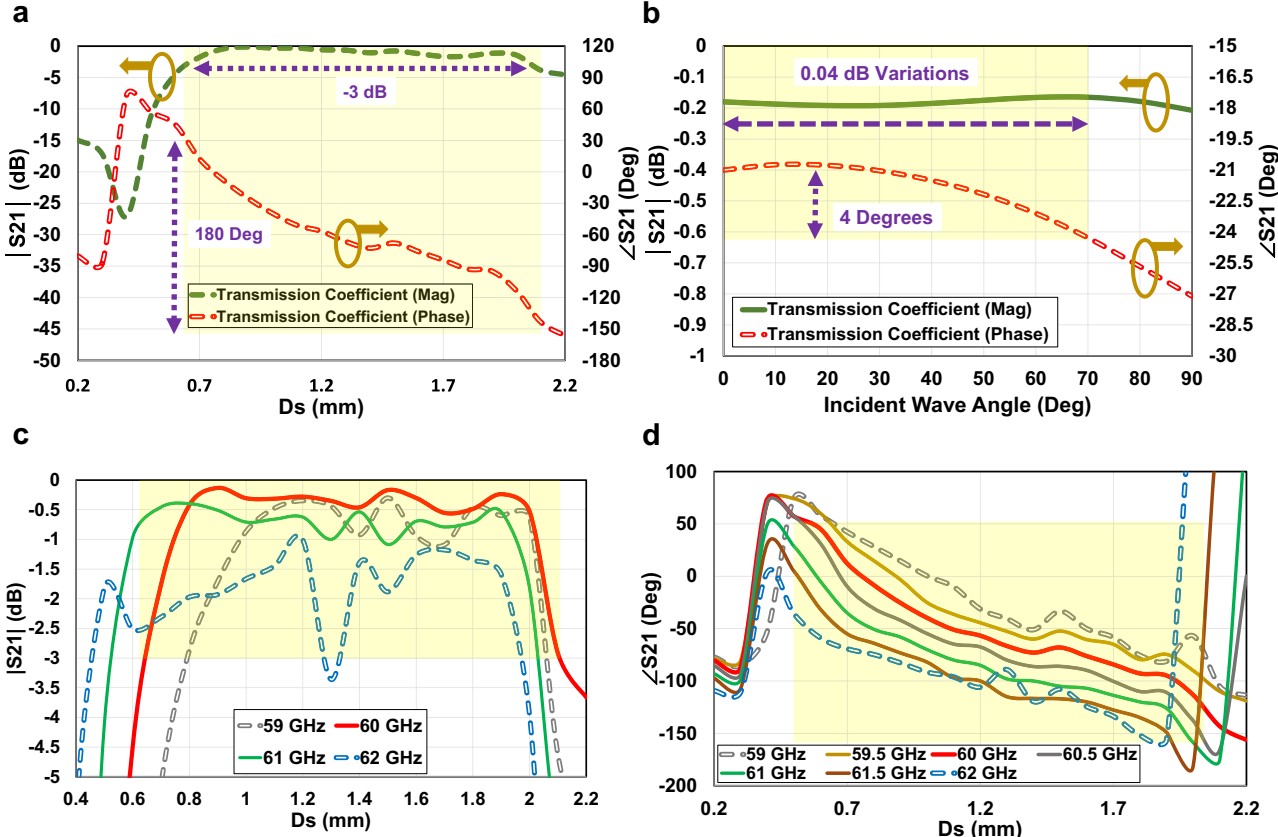

**Fig. 7 | Simulated transmission coefficients (S21 magnitude and phase) of the two-layered crossed-dipole unitcell in presence of skin layer. a** Magnitude (Mag) and phase analysis for various metallic branches lengths at center frequency, $D_S = L_S + W_S/2$; (**b**) magnitude and phase analysis for various incident angles at center frequency; (**c**) magnitude investigation across different frequencies within the desired bandwidth; (**d**) phase investigation across different frequencies within the desired bandwidth.

The designed transmissive metasurface consists of $7 \times 7$ phase-compensated radiating elements with an inter-element distance, $W_S$, along both axes as shown in Fig. 8b integrated with the radar system. According to the Fabry-Perot theory, the distance between the radar system and the designed transmissive metasurface is adjusted to maximize the radiated field, as equal as $\lambda/2$ at 60 GHz. In order to make the work more feasible and practical in the manufacturing and measurement process, the human body skin is modelled as a phantom considering two interleaved cylindrical dielectric slabs, a beaker made by Pyrex glass ($\epsilon_r = 4.7$ and $\sigma = 0.5$ S/m at 60 GHz[59]) filled by pure water ($\epsilon_r = 11.17$ and $\sigma = 36.4$ S/m at 60 GHz[60]) with 5 mm height contacting the metasurface without air gap. It can be shown that the effective characteristics of the new dielectric medium (combination of the Pyrex beaker and pure water) is sufficiently close to the human body skin at 60 GHz.

Figure 8c showcases a fabricated prototype illustrating the integration process of a radar system with a transmitarray metasurface, utilizing a 3D-printed dielectric housing for precise stabilization, while Fig. 8d displays the measurement setup necessary for measuring near-field power density inside the phantom. Some of the parameters shown in Fig. 8 are fixed and selected by the dimensions of the Infineon radar chipset; $W_{SU} = 26$, $L_{SU} = 40$, $W_R = 6.5$, $L_R = 5$ (all in mm).

The proposed structure is simulated using a full-wave electromagnetic simulator, HFSS, and the results of near electric field magnitude and focused power density of the radar system in the presence or absence of the proposed transmitarray metasurface are obtained inside the pure water modelled as a cylindrical slab. For this model, the maximum peak of the power density occurs at a 2 mm distance above the metasurface, 1 mm above the beaker bottom, inside the pure water model. As shown in Fig. 9a, b the designed metasurface provides a significant enhancement on the near electric field

inside the water which is 8.8 times higher than the radar antenna without metasurface. It is expected that the near electric field enhancement provides a very directive near-field power density with high intensity.

Figure 9c, d show the 2D contour plots of the power density radiated by the TX radar antenna in the presence and absence of the near-field-focused planar metasurface in a rectangular region, lying on the plane in parallel to the array with 2 mm distance. It is apparent that using the near-field-focused metasurface provides 11.5 dB (more than 14 times) improvement in the radiated power density across a z-constant surface above the metasurface inside the beaker-filled with pure water model at 60 GHz.

The S-parameter analysis of the simulated reflection and transmission coefficients, S(TX-TX) and S(TX-RX3), for the radar system with and without metasurface, are also investigated, and compared in Fig. 9e. The investigation of the reflection coefficient at the radar TX port shows that using a proper design of the unitcell in contact with the skin model makes an array which is high impedance matched with the beaker filled with water medium at the frequency bandwidth of 59 to 63 GHz. Figure 9e shows that considering the model of the beaker filled with pure water provides 11.5 dB power reflected enhancement from the water medium to one of the radar receivers, RX3, leading to significant enhancement of the radar SNR.

According to the phase analysis presented in the previous section, the microstrip crossed electric dipoles as transmitting elements are properly designed for the compensation of the differential spatial phase delays from the radar antenna radiation. However, in the design of the proposed metasurface two assumptions have been considered: (i) the source antenna should be placed at the center location of the metasurface so that covering the antenna symmetrically; as a consequence, when the metasurface is integrated with the radar system, the TX antenna has an offset from the center of the array causing phase difference between the elements. The

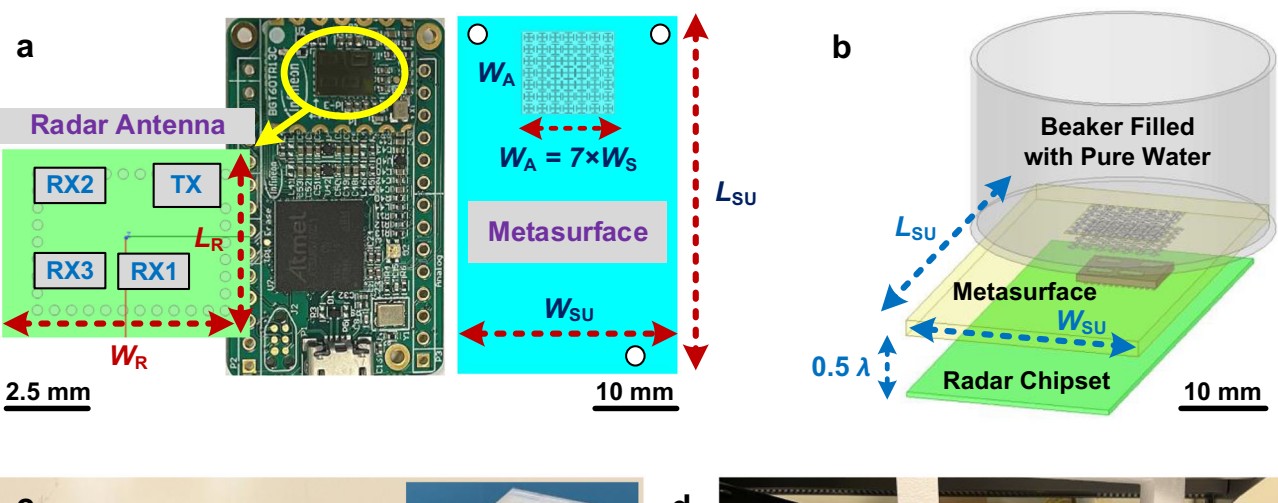

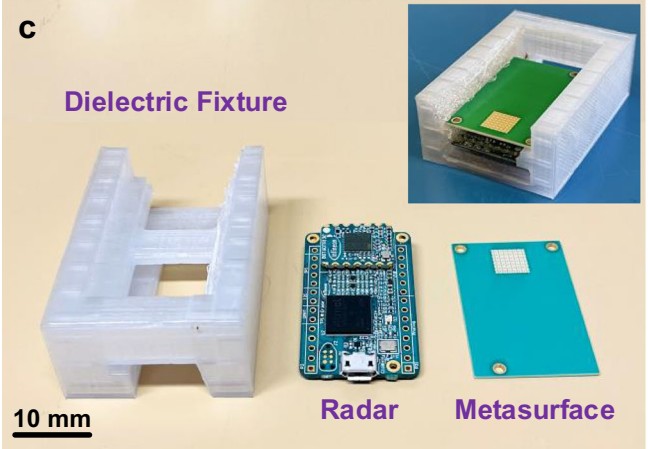

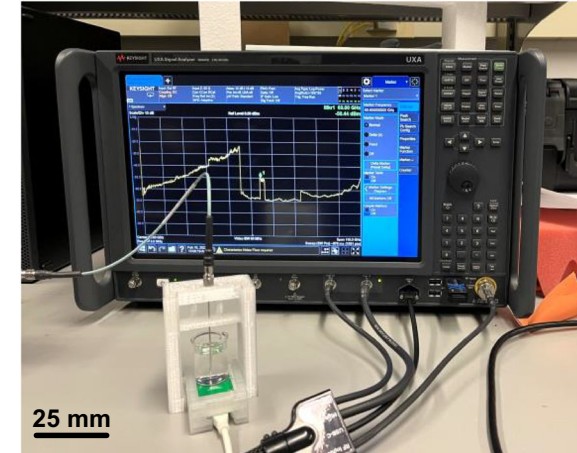

**Fig. 8 | Radar-Metasurface integration for near-field power density measurement using a human skin model phantom. a** The Infineon BGT60TR13C radar chipset with highlighted transmitter (TX) and receiver (RX) antennas and the designed phase-corrected transmitarray metasurface (**b**) radar integrated with the metasurface contacting the proposed phantom, a beaker filled with pure water, in replace with human skin model; (**c**) fabricated prototype showcasing the integration process of a radar system with a transmitarray metasurface, utilizing a 3D-printed dielectric housing for enhanced design precision; (**d**) measurement setup required for measuring near-field power density inside the phantom.

required phase shift compensation when the source is offset and how it differs from the case where the source is positioned in the center are investigated as shown in Fig. 6e, f. As seen in Fig. 6f, when the TX radar antenna is excited, the maximum phase difference of 50° caused by the edge elements affects the array required phase shift leading to phase error loss and E-filed focused deviation. As investigated, this phase error loss causes degradation of 1.2 dB in the maximum accessible focused power density inside the human skin model. Moreover, the offset TX source makes the focused power inside the skin deviate from the orthogonal direction which is not a concern in this application; (ii) The radar antenna radiation illuminates the metasurface unitcells in the normal and oblique directions according to its alignment with the array. In such a case, the phase analysis presented in Figs. 6, 7a are studied based on the normal incident considerations. Nevertheless, the phase shift delays of the normal incident wave illuminated the unitcell is different from the oblique incident one[57]. Although using the proposed symmetric metasurface increases the NF power radiated from the TX antenna and the received power to the RX antenna; the phase error caused by oblique incident radiation from the feed can prevent reaching the maximum power improvement as can be obtained for a microstrip patch antenna placed at the center of the array[25].

## Sensitivity analysis in near-field-focused bio-sensing design

Sensitivity analysis is vital in optimizing the proposed near-field metasurface-enhanced radar for biomedical applications. By discerning pivotal parameters shaping device performance, one can precisely refine the design to attain desired sensitivity and accuracy levels. This process enhances the device's robustness, ensuring consistent functionality across diverse conditions, including varied testing scenarios. Furthermore, sensitivity analysis facilitates precise customization, adapting the radar to meet specific biomedical application requirements.

The first analysis involves an exploration into the critical dynamics of permittivity across inter-individual differences in human skin. The electrical characteristics of human tissues, particularly permittivity and conductivity, demonstrate considerable variability among individuals. The intricate interplay of factors, including hydration levels, age, health status, and tissue composition, adds complexity to the characterization of electrical properties. Permittivity, shaped by tissue composition and structure, tends to exhibit more pronounced variations. In contrast, conductivity, influenced by factors like ion concentration and moisture content, generally shows less noticeable fluctuations.

This paper contributes by introducing a methodology to significantly enhance radar sensitivity for near-field biomedical sensing, particularly in applications such as glucose monitoring. Given the direct contact of the metasurface with the skin, accurate consideration of the permittivity and conductivity values of the skin is crucial. In bio-sensing applications that involve skin, fat, muscle, and bone, high inter-individual differences are common due to variations in body shape. However, for specific applications like glucose monitoring or skin cancer detection, where the 60 GHz signal is

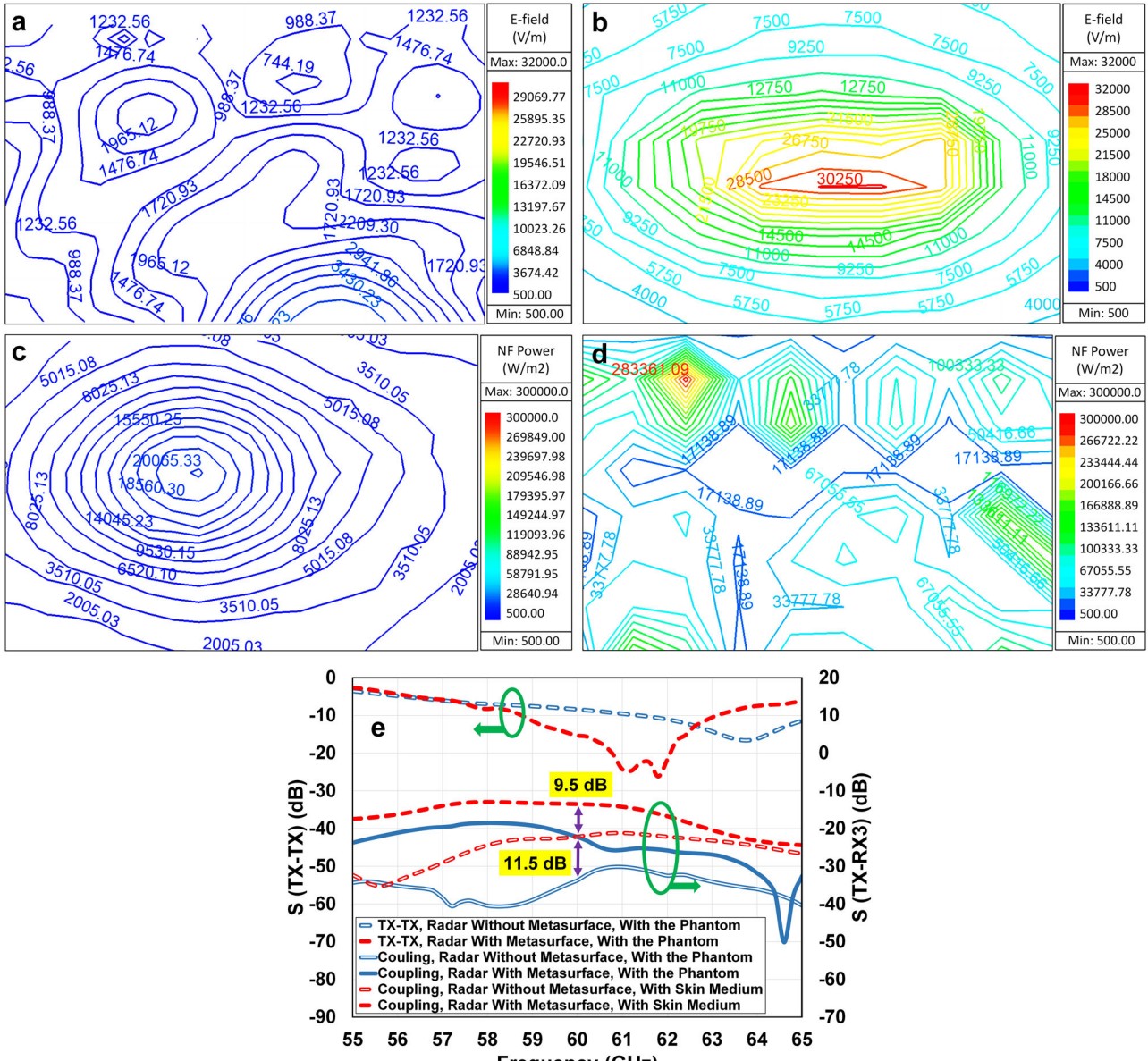

**Fig. 9 | Simulated 2D contour plots of radar system transmitter antenna near-field radiation inside the phantom model at 60 GHz, illustrating on a plane positioned 2 mm above the metasurface. a** Radiated electric field (E-Field) intensity of radar system without metasurface; (**b**) radiated electric field intensity of radar system integrated with the metasurface; (**c**) radiated near-field (NF) power density of radar system without metasurface; (**d**) radiated near-field power density of radar system integrated with the metasurface. **e** The S-parameter analysis, simulated reflection and coupling magnitude of the radar system transmitter (TX) and receiver (RX3) antennas integrated with and without the metasurface in presence of the phantom model.

rapidly attenuated into the body within a few millimeters, individual variations in skin dielectric properties are not substantially impactful. Therefore, the investigation focuses on understanding how a maximum 10% tolerance in permittivity variations influences the metasurface's performance.

Modifying skin permittivity influences the central frequency and the transmission coefficient within the metasurface unitcell analysis. As illustrated in Fig. 10a, at the operational frequency of 60 GHz, a deviation of ± 5% in typical human skin permittivity causes a transmission coefficient reduction of less than 1 dB, while a deviation of ± 10% results in reductions of 2 dB and 1.2 dB, respectively. Extending the evaluation to the array structure, it can be shown that altering permittivity by ± 5% leads to transferred power reductions of 1.4 dB and 0.9 dB. Modifying the permittivity by ± 10% results in changes leading to power reductions of 2.3 dB and 1.9 dB.

Conducting a comparative analysis between the proposed phantom and human skin is the next imperative analysis to gain a comprehensive understanding of how well the phantom mimics the dielectric properties of human skin. The analysis enables the identification of any discrepancies or limitations in the proposed phantom, allowing for refinement and improvement.

The difference between the power reflected from the human skin slab and the proposed phantom, beaker filled with pure water, is investigated in Fig. 9e. As shown, the metasurface in the presence of the skin model provides 9.5 dB enhancement in the reflected power, whereas the enhancement was 11.5 dB when the beaker was filled with pure water. The difference in the material effective permittivity, $\epsilon_e$, as well as the conductivity, $\sigma$, between the human skin model and the beaker filled with pure water accounts for this 2 dB difference in S(TX-RX3) values.

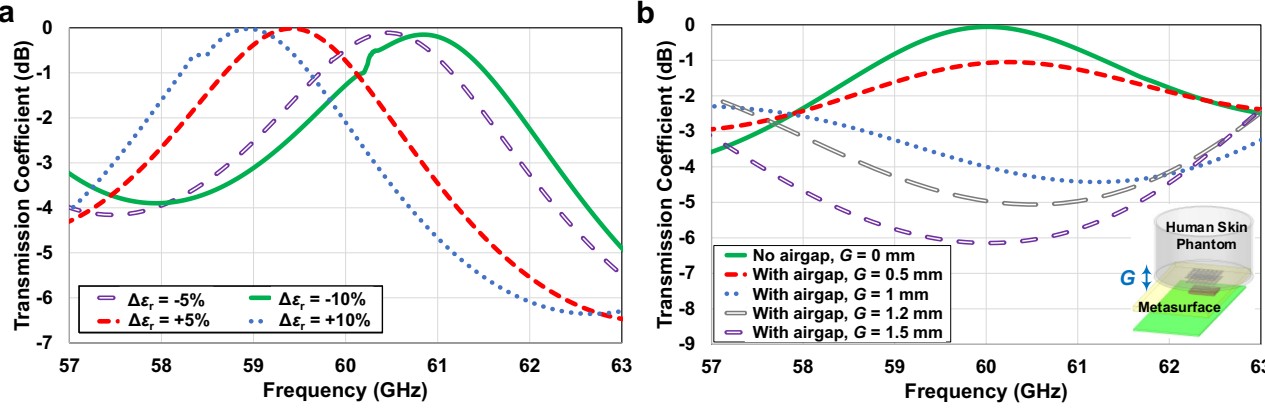

**Fig. 10 | Sensitivity analysis of bio-sensing device considering human skin properties and air gap thickness. a** Investigation of human skin permittivity, $\epsilon_r$, variations and (**b**) air gap thickness between the metasurface and human skin phantom, $G$, variations and their impact on the transmission coefficient of metasurface unitcell.

The permittivity and conductivity of pure water at the frequency of 60 GHz, under typical room temperature conditions, are 11.17 and 65.3 S/m, respectively[60]. Consequently, it becomes imperative to consider the impact of the external shell in the calculations. According to the literature, Pyrex glass demonstrates a relative permittivity of 4.7 and a conductivity of 0.5 S/m, at 60 GHz[59]. In such situations, one can compute the effective dielectric properties of composite materials arranged in series across the entire volume facing the metasurface[61].

The effective permittivity of the overall medium, a beaker filled with pure water, is determined to be 8.75. Consequently, comparing the permittivity of the proposed phantom, 8.75, to that of a typical human skin model, 7.98, at 60 GHz reveals a difference of approximately 0.77. When comparing the permittivity of the phantom with that of human skin, it is noted that the tolerance is within +10%. Consequently, as investigated in Fig. 10a, the anticipated outcome is a 2 dB reduction in the power transferred into the phantom compared to the power transferred into human skin. In this scenario, it can be inferred that the difference in focused power has been transformed into dissipated and reflected power, contributing to an enhancement in the reflection power when using the phantom as opposed to the human skin slab. On the conductivity front, the calculated conductivity of the phantom (a beaker filled with pure water), at 60 GHz and room temperature stands at 42 S/m. In contrast, the conductivity of human skin at 60 GHz is measured at 36.4 S/m. This implies that the phantom medium serves as a marginally better conductor, resulting in increased power reflection directed towards the radar.

In conclusion, the examination of both permittivity and conductivity indicates that utilizing the proposed phantom, a beaker filled with pure water, results in less transferred power into the medium compared to the human skin medium, leading to an enhancement in power reflection. This is evident in Fig. 9e, where the power reflection from the phantom (11.5 dB) is 2 dB higher than the power reflection from the skin medium (9.5 dB).

In the last phase of sensitivity analyses, the impact of human skin non-uniformity on planar transmissive metasurface performance is investigated. The proposed metasurface is designed as a planar interface to be used with a rigid structure; however, the human body deviate from complete planarity, introducing the possibility of an air gap between the metasurface and the skin. This metasurface-enhanced radar near-field sensing method exhibits versatility across biomedical applications, including glucose monitoring, skin cancer detection, and on-body radar cardiorespiratory monitoring. The study emphasizes the development of a highly sensitive millimeter-wave radar for real-time disease diagnostics, with a specific focus on continuous blood glucose monitoring for diabetic care. The primary objective is to integrate glucose monitoring seamlessly into wearable devices, as depicted in Fig. 1, tailoring the metasurface technology for this targeted application context.

In the context of utilizing metasurface-enhanced radar for wrist-worn wearable devices, the effective area of the skin is treated as planar, assuming minimal or no gap between the metasurface and the body. This is supported by the small effective area of the metasurface ($7.7 \times 7.7$ mm$^2$) and the secure fit of devices like smartwatches, ensuring practical biomedical sensing and achieving high measurement accuracy. However, it is essential to explore the performance of the metasurface under conditions where a significant gap exists between the metasurface and the human skin, particularly relevant in other biomedical contexts such as cancer detection. This consideration arises from the fact that in certain applications, there may be a notable separation between the designed metasurface and the human body, warranting an analysis of the impact on the metasurface performance under such conditions.

The simulation analysis encompasses the transmission coefficient of the metasurface unitcell and the transmitted power enhancement of the array structure, considering varying air gap thicknesses from 0 to 1.5 mm. Illustrated in Fig. 10b is the unitcell analysis under different air gap conditions. The findings reveal that a 0.5 mm air gap induces a 1 dB reduction in the transmission coupling factor crucial for effective power transfer. The introduction of a 1 mm air gap causes a notable impedance mismatch, resulting in a 4 dB reduction in transmitted power due to the air acting as an additional load. Successive increases in the air gap led to further reductions in the transmitted power and an increase in the power reflected from the air-skin interface. Extending the investigation to analyze transmitted power enhancement using the metasurface array within the skin medium, while considering air gap distances of 0.5, 1, and 1.5 mm, demonstrates a decrease in near-field power enhancement inside the skin of 1.2, 4.5, and 7.5 dB, respectively. It is crucial to emphasize that the designated near-field focal point, positioned 2 mm above the metasurface, does not penetrate the skin medium in the presence of an air gap measuring 2 mm or more. Consequently, this non-penetration leads to a lack of observed power enhancement. Under these conditions, the outcomes align with those observed when the metasurface is not integrated.

In scenarios where uniformity is pivotal for deploying the presented metasurface-enhanced radar system in various biomedical applications, several strategies can mitigate the impact of non-uniformity. Firstly, reducing the metasurface array size while preserving resolution enhancement capability addresses non-uniformity concerns by maintaining consistent results in a smaller area. Secondly, incorporating a flexible substrate in the metasurface design allows it to conform to body skin contours, effectively minimizing non-uniformity issues. Thirdly, employing machine learning algorithms, coupled with signal filtering techniques, robustly accounts for skin variability in radar power reception, mitigating interference. Lastly, creating a wideband metasurface enables the radar system to operate across frequencies, offering advantages in penetrating the skin at different depths

and interacting distinctively with skin features, potentially providing a more comprehensive perspective.

### Near-field transmitted power density and radar SNR measurement

In the fabrication process detailed in Supplementary Note 1, the prototype of the transmitarray metasurface, as depicted in Fig. 8c, showcases its integration with the radar module. This integration is achieved using a 3D printed dielectric fixture, strategically employed to stabilize the metasurface at a half-wavelength air gap distance above the radar system. In the following, the two sets of measurement processes are discussed. To ensure accurate results, the signal processing configuration for the FMCW Infineon radar system operating at 60 GHz is presented in Supplementary Note 2 and considered in the measurement process, utilizing Supplementary Fig. S1 and Supplementary Table S1.

As shown in Fig. 8d, a special probe working at 60 GHz is connected to a spectrum analyzer supporting the frequency range up to 110 GHz and is used for the power measurement. A Pyrex beaker filled with 10 ml pure water is utilized to be placed on top of the metasurface array with a half-wavelength, 2.5 mm, distance above the radar antenna. The experiment is repeated for the radar system without a metasurface such that very thin cardboard is replaced with the metasurface at the exact location to maintain the beaker at a specified distance above the radar surface. In both cases, the probe is immersed into the beaker with a 1-mm distance above the beaker's bottom surface, which is 2 mm above the metasurface layer location.

The measurement results showing the power transferred into the water medium and picked up by the probe are presented and compared in Fig. 11a. As can be seen, using the designed transmitarray metasurface enhances the power received by the probe at the operational radar frequency range, from 59.7–61.7 GHz. Fluctuations in the enhanced transferred power result from both the distribution of radar power and specific design considerations. Looking specifically at the designed frequency of 60 GHz shows an improvement of 11 dB in the measured near-field power, which is in good

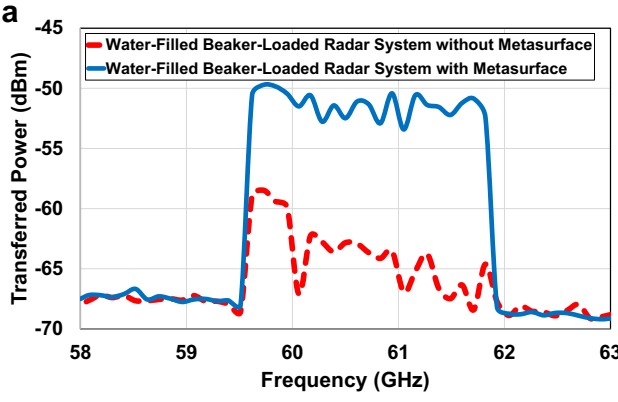

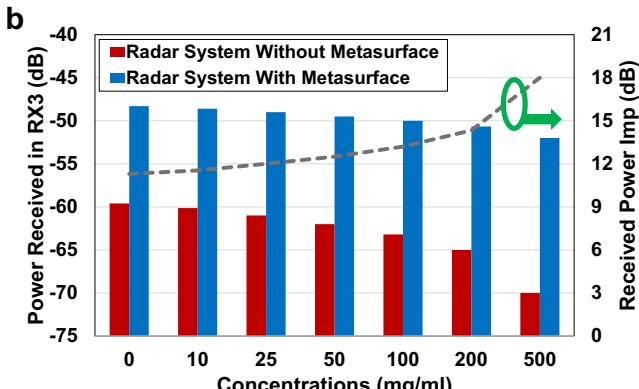

**Fig. 11 | Measurement results depicting the transmitted and received power of the metasurface-enhanced radar system in presence of the proposed phantom. a** The measured near-field power density incident by Infineon radar transmitter (TX) antenna inside the phantom, beaker filled with pure water, in the presence or absence of the designed transmissive metasurface. **b** The investigation of the measured power reflected from the phantom, received by the radar receiver antenna (RX3), over a period in the presence or absence of the designed transmissive metasurface for different glucose concentration levels.

### Table 1 | Features and advantages of utilizing metasurface-enhanced mm-wave radar for biomedical sensing

| Features | Descriptions |
|---|---|
| Near-Field Theory Considerations | The metasurface is designed with a methodology tailored for near-field radiation theory, essential for biomedical sensing, in contrast to prior works focusing on usual far-field antenna radiation. |
| Compactness | In near-field focusing, current antenna designs face integration challenges with low-profile radar due to their bulky structures. This paper proposes a low-profile, compact, and planar metasurface for seamless integration with a radar system. |
| Millimeter-wave Operation | The mm-wave frequency range is selected for its high resolution and precision, bio-compatibility, and reduced interference making it suitable for applications in wearable devices like smartwatches. |
| Direct Human Skin Contact | Near-field studies typically focus on antenna radiation in free space, but the human body causes detuning and impedance mismatching, degrading performance. Existing methods are unsuitable for direct-contact biomedical applications. The designed metasurface allows skin contact, enabling precise targeting without interference. |
| Impedance Matching Layer | The metasurface is carefully designed based on impedance matching network theory to achieve highly effective impedance matching between free space and the human skin. This design tackles a significant challenge in the literature by minimizing reflections arising from air-skin interference. |
| Intensify Radiated Near Electric Field | The metasurface, with phase-synthesized unicells significantly boosts the near-field electric field from the source antenna, achieving an 8.8-fold improvement within the proposed skin phantom medium compared to the radar antenna without the metasurface. |
| Intensify Near-field Transmitted Power | The metasurface enhances absorbed power in human skin at 60 GHz without affecting source antenna impedance matching. This near-field-focused design yields an 11 dB improvement in radiated power density above the metasurface within the proposed skin phantom medium. |
| Intensify Near-field Reflected Power | Analyzing the radar's reflection coefficient shows the metasurface amplifying reflected power from human skin by 11.3 dB. This enhancement leads to a notable 13.4-times improvement in radar SNR, enabling effective information transmission and high-performance near-field sensing in biomedical applications. |
| Glucose Sensing Enhancement | In glucose monitoring, metasurface-enhanced power measurements show a resolution of around 0.5 dB per 15 mg/ml glucose concentration, maintaining a high SNR and confirming enhanced functionality in non-invasive glucose sensing applications. |

agreement with the simulation results representing the power density improvement of 11.5 dB in Fig. 9c, d.

The next step is the radar SNR investigation by measuring the reflected power from the phantom to one of the radar receiver antennas (RX3). The Supplementary Algorithm S1 is used to measure the received power in a period of time in the cases of the radar system loaded by a beaker filled with pure water in the presence or absence of the transmitarray metasurface. To illustrate near-field sensing improvement with higher SNR, different concentration levels of glucose are added to the pure water and the results are analyzed and compared in Fig. 11b. As shown, using the near-field-focused metasurface enhances the power reflection from the phantom by 11.3 dB and improves the radar SNR around 13.4 times, which is in good agreement with the simulation results presented in Fig. 9e. Furthermore, in this experimental study, the effect of varying the dissolved amount of glucose in water in the presence of the metasurface is explored. 200 experiments for each scenario were carried out to provide high repeatability. Although power reflection is reduced by increasing the glucose concentration in both cases, the metasurface clearly improves the received power level by the radar as shown in Fig. 11b. The received power in the presence of the metasurface for different glucose concentration levels has a resolution of around 0.5 dB per 15 mg/ml concentration, validating that the metasurface enhances the overall sensing functionality. Table 1 provides a comparison informative descriptions to thoroughly discuss and emphasize the novelty and advantages of the proposed method using the designed metasurface.

## Conclusions

On-body radar sensors represent a crucial advancement in biomedical sensing technology, enabling continuous, real-time monitoring of vital signs, glucose levels, and health metrics that can provide early diagnosis, improve treatment, and ultimately save lives. This paper introduces a low-profile planar metasurface which is specifically designed with impedance matching capabilities, seamlessly integrated with radar transmitter and receiver antennas, facilitating direct contact with a human body simplified model to enable substantial advancements in near-field sensing performance for biomedical applications. The key aspects contributing to the metasurface's remarkable radar near-field sensing were the enhanced power density absorption from the radar antenna transmitter into a controlled medium alongside the elevated received power level by the radar antenna receiver, leading to a higher system signal-to-noise ratio. Specifically, inside the simplified liquid container, the use of the metasurface led to an improvement of more than 11 dB in the near-field Poynting power density. Moreover, through radar signal processing, the analysis revealed an additional improvement of over 11 dB in radar signal-to-noise ratio, thereby enhancing the sensor sensing abilities.

## Data availability

The data that support the findings of this study are available from the corresponding author upon reasonable request. The data used in the metasurface design can be simulated by means of the full-wave electromagnetic software with the properties explained in the manuscript.

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

## Acknowledgements

The authors gratefully acknowledge the support received from NSERC, MITACS, OCI, CMC, Ansys, Infineon, and Google. This work would not have been made possible without support from Infineon and Google. The authors sincerely thank Dr. Jaime Lien for the pivotal role in enabling this project to become a reality.

## Author contribution

M.O.B. conceptualized the idea, developed the methods, conducted theoretical analyses, designed simulations, executed fabrication and experiments, and took the lead in drafting and extensively revising the manuscript. A.G. collaborated on performing the experiments, interpreting the results, validating the findings, and made partial contributions to the writing process. S.A. contributed to building the measurement setup, executing fabrication, and conducting experiments. O.R. provided guidance on theory and made partial contributions to the writing and revision process. G.S. contributed to conceptualizing the idea and designing the experiments, made contributions to the writing and revision process, and supervised all aspects of the project.

## Competing interests

The authors declare no competing interests.

## Additional information

**Supplementary information** The online version contains
Supplementary Material available at

Mohammad Omid Bagheri.

**Peer review information** *Communications Engineering* thanks the
anonymous reviewers for their contribution to the peer review of this work.
Primary Handling Editors: Anastasiia Vasylchenkova, Rosamund Daw. A
peer review file is available.

