## [Peer Review File · Communications Engineering]

Reviewers' comments:

Reviewer #1 (Remarks to the Author):

The topic discussed in this paper is of interest of the community, but the following major concerns need to be addressed.

1) The proposed metasurface is planar and it is supposed to be used in combination with a rigid structure. However, even if the human body can be approximated with a planar interface, the body surface is not completely planar. It is likely that in many applications, an air gap may appear in between the metasurface and the skin. To estimate the effective performances in realistic case applications, it is then important to add this effect and evaluate its impact at least in the numerical analysis.

2) In the manuscript the authors consider the typical permittivity value for human skin at 60 GHz. However, given the inter-individual differences, this value may vary. How does this impact the performance of the metasurface?

3) The phantom proposed for the work has an external shell. In this frequency range, this typically affects the phantom's characteristics. The authors state that "It can be shown that the effective characteristics of the new dielectric medium (combination of the Pyrex beaker and pure water) is sufficiently close to the human body skin at 60 GHz.", but a characterization of the permittivity is lacking. Also, the authors need to quantify how much the permittivity of the phantom differs from the skin one at 60 GHz.

4) The authors state that the lack of uniformity is justified by the application, but it is not clear what is the application or at least a selection of possible applications of this technology. Depending on this answer the non-uniformity can be more or less tolerated.

5) Could the authors comment on possible solutions to this lack of uniformity?

6) Together with the metasurface, the radar system is presented with a detection algorithm. The algorithm is pretty simplistic. How does the system effectively performs when measuring one or multiple targets inside the skin? How well can they be located with and without the addition of the metasurface?

7) There are several typos in the text and the symbols used for the equations (e.g., different symbols to indicate the phase shift ϕ).

Reviewer #2 (Remarks to the Author):

The paper is well-written and effectively communicates the research findings, contributing significantly to the existing body of knowledge in the field.

Comments to be addressed:

Has the suggested unit cell been previously introduced? What distinguishes it in terms of novelty and

advantages when compared to other existing studies?

It's crucial to delve into the design intricacies and features of the 4-element radar antenna.

Furthermore, there needs to be an in-depth discussion about the distance between the source antenna and the metasurface layer.

Additionally, it is imperative to include a comparison table with informative descriptions to thoroughly discuss and emphasize the novelty and advantages of the proposed method and the designed metasurface.

Furthermore, the references should be updated to incorporate more recently published papers, preferably from 2023, for a comprehensive and current overview of the field.

Reviewer #3 (Remarks to the Author):

In this work the metasurface is integrated with a mm-wave radar chipset from Infineon and a human skin model to improve the radar performance.

Authors demonstrate that their metasurface study enhances the power density absorption and reflection, as well as the signal-to-noise ratio, of the radar system when in contact with a human body model.

The paper is written well, however it lacks biomedical related content. It mainly focuses on study of antenna design and the impact of metasurface study.

Some specific comments:

- In Figure 3, the simulations indicate that the ports are placed on both ends of the model, like a transmittance model. Whereas in the measured scenario based on Fig. 7, the receiver antenna RX3 and transmitting antenna TX are on the same end. Would the results be the same as that of the earlier simulation?
- Could you elaborate on the significance of crossed dipole unit cell? How do you compare them with literature? Are there any additional advantages?
- Are there any effects from antenna radiation pattern, directivity, and bandwidth on the near-field sensing performance?

Response Document: Radar Near-Field Sensing Using Metasurface for Biomedical Applications

Manuscript ID: COMMS-23-0421-T

Authors: Mohammad Omid Bagheri, Ali Gharamohammadi, Serene Abu-Sardana, Omar M Ramahi, and George Shaker

Dear Editor and Reviewers:

We want to express our sincere appreciation for the meticulous evaluation of our manuscript (Paper No.: COMMS-23-0421-T) entitled “Radar Near-Field Sensing Using Metasurface for Biomedical Applications”. Your valuable insights and suggestions are greatly appreciated. We have carefully implemented the necessary revisions, which are now highlighted in red within the updated manuscript. In this response letter, we have addressed your comments and suggestions in black, while our corresponding responses and the modifications made are indicated in blue text. We kindly request your attention to the detailed description provided below.

Comments from Reviewer 1:

The topic discussed in this paper is of interest of the community, but the following major concerns need to be addressed.

Dear Reviewer,

We express our sincere appreciation for your considerate assessment of our manuscript. Your feedback undoubtedly has contributed to enhancing the quality and clarity of our manuscript. We acknowledge your valuable suggestions for improvement and the identification of areas that require correction. We have proofread the manuscript and addressed all of the issues that you have pointed out.

1. The proposed metasurface is planar and it is supposed to be used in combination with a rigid structure. However, even if the human body can be approximated with a planar interface, the body surface is not completely planar. It is likely that in many applications, an air gap may appear in between the metasurface and the skin. To estimate the effective performances in realistic case applications, it is then important to add this effect and evaluate its impact at least in the numerical analysis.

Authors' Reply:

In response to this comment, the impact of the air gap is discussed by providing some simulation results. However, first, we aim to outline reasons explaining why non-planarity may not significantly affect the metasurface performance for a specific application within this paper.

- 1) **Considering the Specific Application:** The methodology presented, metasurface-enhanced radar near-field sensing, demonstrates broad applicability in diverse biomedical sensing contexts including glucose monitoring [2], [15], [16], skin cancer [18], as well as heart and on-body radar cardiorespiratory monitoring [20]. Our research group has been actively engaged in diverse fields related to biosensing applications. In a recent study, [20] introduced a chest-wearable 60 GHz radar system for continuous monitoring of cardiorespiratory displacement waveforms. In [2], a novel

design of a portable planar microwave sensor operating within the ISM band 2.4–2.5 GHz is introduced, enabling rapid, accurate, and non-invasive blood glucose level monitoring. [15] introduces an integrated millimeter-wave radar system to detect different glucose concentration levels in artificial blood samples. This study aims to affirm the suitability of mm-wave radars for non-invasively monitoring diabetes patients' glucose levels, leveraging signal processing approaches to identify various glucose concentrations and correlate them with the reflected mm-wave readings. In the work detailed in [16], a novel approach to monitoring glucose levels is presented, utilizing a robust low-power millimeter-wave radar system to differentiate between blood samples with varying glucose concentrations. [22] proposes an alternative glucose monitoring approach using an antenna sensor operating at 4.2 GHz positioned on the pancreas to capture dielectric radiation signals. [23] details the development of a novel microwave biosensor for real-time non-invasive glucose monitoring. This biosensor incorporates a rectangular plexiglass channel integrated into a triple-pole complementary split ring resonator, operating in the centimeter-wave range of 1–6 GHz with a sensitivity performance of approximately 6.2 dB/(mg/ml) for the sensor.

Figure 1 (a) A wearable device equipped with radar technology positioned near a human hand; (b) Incorporating a metasurface structure between the radar antennas and the hand to enhance the near-field energy coupling and signal-to-noise ratio (SNR).

This paper specifically emphasizes the potential of realizing a highly sensitive mm-wave radar for real-time disease diagnostics, focusing on the vital role of continuously monitoring blood glucose concentrations for monitoring diabetic patients. The ultimate aim of this work is to integrate glucose monitoring capabilities into wearable devices as shown in Fig. 1, such as smartwatches, emphasizing the specific application context for which this metasurface technology is tailored. The FMCW radar (Infineon, BGT60TR13C) with a 60-GHz center frequency is selected, which our group successfully utilized previously for near-field probing of human tissue, particularly, glucose level detection [2], [20]. As presented in the manuscript, Fig. 14b shows the analysis of the power received by the radar RX antenna from the proposed phantom (beaker filled with pure water) in the presence or absence of the designed transmissive metasurface for different concentration levels of sugar.

Figure 14(b) The investigation of the power reflected from the beaker filled with pure water received by RX3 radar antenna over a period in the presence or absence of the designed transmissive metasurface for different concentration levels of sugar.

The introduction section of the manuscript has been updated (highlighted in red) by adding Fig. 1 and explaining the application related to this work.

- Given a specific application, using metasurface-enhanced radar as a wearable device on the wrist, makes it evident that there would be no gap between the metasurface and human skin. This is because the total dimension of the metasurface effective area is $7.7 \times 7.7 \text{ mm}^2$, while according to the literature, the average male wrist circumference is 6.89 inches (17.5 cm), and the average woman's wrist circumference is 5.9 inches (15 cm). If we take into account the smallest wrist circumference, the portion of the effective area in contact with a wearable device like a smartwatch is approximately 6.5 cm. As illustrated in the figure below, the metasurface array's area in contact with the hand is about 9 times smaller than the total area. Moreover, devices such as smartwatches can be worn to fit tightly around the wrist, leaving no gaps. Consequently, we can treat the effective area as planar, assuming no gap between the body and metasurface with a high degree of accuracy. Additionally, the compact effective area of the metasurface makes it suitable for biomedical sensing on various body parts.

- Given that there is generally no substantial gap between the designed metasurface and human body skin, it is also of interest to analyze the performance of the metasurface if there is a gap to the human skin in rare cases. The simulation results involve the radar integrated with a metasurface, considering air gap variations from 0 to 1.5 mm. The simulation analysis encompasses the transmission

coefficient of the metasurface unit cell and the transmitted power enhancement of the array structure.

The graph below depicts the unit cell analysis considering various air gaps. The investigation demonstrated that a 0.5 mm air gap resulted in a 1 dB reduction in the transmission coupling factor necessary for the effective transfer of high power by the metasurface. Expanding the gap by 1 mm resulted in a more significant impedance mismatch in the near field. This occurred because the 1 mm thick air gap acted as an additional load between the metasurface and the human skin, leading to a 4 dB reduction in transmitted power. The introduction of additional air gaps resulted in further reductions in transmitted power and an increase in power reflected from the air-skin interface.

Analysis of the transmitted power enhancement using metasurface into the skin medium considering air gaps is shown in the figures below. It is important to emphasize that the power is consistently measured at a distance of 2 mm above the metasurface at the specified focal point defined in the metasurface design for all scenarios.

In general, it is expected that increasing the gap between the metasurface and human skin medium causes impedance mismatching between the two mediums leading to increased reflection power from the interface and decreased focused transmitted power into the skin. On the other hand, the findings demonstrate that the metasurface continues to enhance the focused power even as the gap extends to 2 mm. The near-field power enhancement values corresponding to the different air gaps are presented in the table below. It is important to highlight that in the design, the designated near-field focal point is situated 2 mm above the metasurface. Consequently, when there is an air gap of 2 mm or more, the focal point does not penetrate the skin medium, resulting in no power enhancement. In such cases, the outcomes align with those observed when the metasurface is not integrated. A paragraph explaining sensitivity analysis including the possibility of the air gap is presented in the manuscript, at the end of the section “Near-field analysis of the radar system integrated with and without metasurface” which is highlighted in red.

Table A. Air Gap analysis.

The airgap between the metasurface and skin (mm)	Power Enhancement (dB)
No Airgap	11.5
0.5	10.3
1	7
1.5	4
2	2.2

Table B. Power distribution at 2mm cross section above the metasurface.

2. In the manuscript the authors consider the typical permittivity value for human skin at 60 GHz. However, given the inter-individual differences, this value may vary. How does this impact the performance of the metasurface?

Authors' Reply:

As presented, one of the contributions of this paper is to present a methodology to significantly enhance radar sensitivity for near field biomedical sensing, with applications such as glucose monitoring [2], [15], [16], [23]. Since the metasurface is in direct contact with the human body's skin, it is essential to consider the correct values of the permittivity and conductivity of the skin. As reported in [53], the human skin has a typical permittivity value of 7.98 and conductivity of 36.4 S/m at the frequency of 60 GHz. In general, for bio-sensing applications considering skin along with fat, muscle, etc, the inter-individual differences are high due to the body shape differences. However, for applications such as glucose monitoring or skin cancer detection, it is well documented that the 60GHz signal will be rapidly attenuated into the body (within few mm) [2], [15], [16], [23]. Therefore, when accounting for individual variations in skin dielectric properties, the changes in permittivity at 60 GHz are not substantial. Thus, our investigation focused on understanding how a maximum of 10% tolerance in permittivity variations impacts the performance of the metasurface.

The figure below illustrates the examination of the transmission coefficient (S_{12}) within the metasurface unit cell. It demonstrates that altering the skin's permittivity leads to variations in the central frequency and a decrease in the transmission coefficient at the frequency of interest. At 60 GHz, a deviation of $\pm 5\%$ results in a reduction of S_{12} by less than 1 dB, while a deviation of $\pm 10\%$ causes reductions of 2 dB and 1.2 dB, respectively.

The evaluation extends to the array structure, examining how variations in permittivity affect the transferred power density into the human skin. This investigation is depicted in the figures below.

Upon the initial analysis focusing on the skin layer with a precise permittivity of 7.98, it was observed that the metasurface has the capability to augment the power within the skin medium by 11.5 dB. The subsequent findings illustrate that altering the permittivity by $\pm 5\%$ leads to a reduction in transmitted power, resulting in decreases of 1.4 dB and 0.9 dB, respectively. Additionally, modifying the permittivity by $\pm 10\%$ results in changes leading to decreases of 2.3 dB and 1.9 dB, respectively.

A paragraph explaining sensitivity analysis including the possibility of the skin permittivity variations is presented in the manuscript, at the end of the section “Near-field analysis of the radar system integrated with and without metasurface” which is highlighted in red.

Table C. Impact of Skin Variations

3. The phantom proposed for the work has an external shell. In this frequency range, this typically affects the phantom's characteristics. The authors state that "It can be shown that the effective characteristics of the new dielectric medium (combination of the Pyrex beaker and pure water) is sufficiently close to the human body skin at 60 GHz.", but a characterization of the permittivity is lacking. Also, the authors need to quantify how much the permittivity of the phantom differs from the skin one at 60 GHz.

Authors' Reply:

This observation is entirely accurate; within this frequency range, the external shell (Pyrex Glass) does indeed influence the characteristics of the phantom. Although we initially decided to simplify the presentation of the methodology, this consideration is now explicitly addressed in the revised manuscript, with the updated parts highlighted in red in the paragraph below.

“In order to make the work more feasible and practical in the manufacturing and measurement process, the human body skin is modelled as two interleaved cylindrical dielectric slabs, a beaker made by Pyrex glass ($\epsilon_r=4.7$ at 60 GHz [59]) filled by pure water ($\epsilon_r=11.17$ at 60 GHz [60]) with 5 mm height contacting the metasurface without air gap. It can be shown that the effective characteristics of the new dielectric

medium (combination of the Pyrex beaker and pure water) are sufficiently close to the human body skin at 60 GHz.”

As per the literature, human skin exhibits a permittivity of 7.98 at the frequency of 60 GHz [53]. However, the permittivity of pure water at this frequency, under typical room temperature conditions, is 11.17. Consequently, it becomes imperative to consider the impact of the external shell in the calculations. According to the literature [59], at 60 GHz, glass demonstrates a relative permittivity of 4.7 and a conductivity of 0.5 S/m. In such scenarios, one can perform calculations to determine the effective permittivity of the entire volume that the metasurface is facing.

There are two commonly used models for calculating the effective permittivity of composite materials, the series model and the parallel model. In the series model, the composite material is treated as a series of layers or capacitors, where the electric field passes through each layer sequentially which is related to our work. In this case, a composite material consisting of two phases, Phase 1 and Phase 2, with respective volume fractions V_1 and V_2 is considered. The ϵ_1 and ϵ_2 represent the permittivities of Phase 1 and Phase 2, respectively. The total electric field E across the composite material is the sum of the electric fields in each phase: $E = E_1 + E_2$.

Applying the relationship between electric displacement D and electric field E ($D = \epsilon E$) to each phase, we have $D_1 = \epsilon_1 E_1$ and $D_2 = \epsilon_2 E_2$. Since the electric displacement D is the sum of the electric displacements in each phase, we can write $D = D_1 + D_2$. Using the relationship $D = \epsilon E$, we can express the total electric field E as $E = D / \epsilon$. Substituting the expressions for D_1 and D_2 , we obtain $E = (D_1 + D_2) / \epsilon$.

Now, we need to use D_1 and D_2 in terms of the respective electric fields E_1 and E_2 as expressed above. Substituting these expressions into the equation for E , we have $E = (\epsilon_1 E_1 + \epsilon_2 E_2) / \epsilon$. Dividing both sides by the total electric field E , we get $1 = (\epsilon_1 E_1 / E) + (\epsilon_2 E_2 / E)$. Recognizing that E_1/E represents the volume fraction V_1 and E_2/E represents the volume fraction V_2 , the formula for the effective permittivity in the series model is calculated by:

$$\frac{1}{\epsilon_r} = \left(\frac{V_1}{\epsilon_1}\right) + \left(\frac{V_2}{\epsilon_2}\right) + \dots + \left(\frac{V_n}{\epsilon_n}\right)$$

Here, V_1, V_2, \dots, V_n are the volume fractions of the individual components, and $\epsilon_1, \epsilon_2, \dots, \epsilon_n$ are their respective permittivities. In this model, the inverse of the effective permittivity is equal to the sum of the volume fractions divided by the permittivities of the individual components.

Using the formula as presented above, one can calculate the effective permittivity of pure water in combination with a Pyrex glass beaker. As mentioned in the manuscript “a beaker made by Pyrex ($\epsilon_r=4.7$) filled by pure water ($\epsilon_r=11.17$) with 5 mm height contacting the metasurface without airgap”. Given that the thickness at the bottom of the glass beaker is measured as 1 mm, the volume fractions are calculated as 0.2 for the beaker (1 mm/5 mm = 0.2) and 0.8 for the pure water (4 mm/5 mm = 0.8). Utilizing the previously mentioned formula, the effective permittivity of the medium is determined as $\epsilon_r(\text{effective}) = 8.75$. Consequently, a comparison between the permittivities of the proposed phantom (8.75) and human skin (7.98) at 60 GHz reveals a difference of approximately 0.77, which is not deemed significant. However, it is important to note that this difference does impact the performance of the metasurface, as previously explained in the manuscript and illustrated in Fig. 11. Fig. 11 illustrates the comparison between the power reflection from the human skin layer and the power reflection from the beaker filled with pure water. The

distinct permittivity and conductivity characteristics of the two mediums result in a notable 2 dB difference in the reported reflection power, with 9.5 dB for the skin and 11.5 dB for the beaker filled with water medium.

The manuscript has been revised accordingly, with the updates highlighted in red as per the explanations provided above.

Figure 11. The S-parameter analysis, reflection and coupling magnitude, of the Infineon radar system integrated with and without the transmissive metasurface in the presence of the beaker filled with pure water.

4. The authors state that the lack of uniformity is justified by the application, but it is not clear what is the application or at least a selection of possible applications of this technology. Depending on this answer the non-uniformity can be more or less tolerated.

Authors' Reply:

The level of tolerance for non-uniformity indeed depends on the intended applications. To address this concern, we elaborated on the potential applications of the technology, providing a comprehensive range of possible uses as referred to in question 1 answer. The introduction section of the manuscript also has been updated (highlighted in red) by adding Fig. 1 and explaining the application related to this work. According to that, non-uniformity may not significantly affect the metasurface performance for a specific application within this paper.

Possible Applications:

The proposed metasurface-enhanced radar near-field sensing methodology finds versatile applications in biomedical contexts like skin cancer detection [18], glucose monitoring [2], [15], [16], and on-body radar cardiorespiratory monitoring [20]. Despite challenges in non-invasive blood glucose level monitoring, especially in achieving high sensitivity and resolution, radar technology, examined through diverse approaches, shows significant potential.

This paper underscores the potential of high-sensitivity mm-wave radar integrated with metasurface for real-time disease diagnostics, particularly in continuously monitoring blood glucose levels for diagnosing

diabetes. The overarching goal is to integrate glucose monitoring into wearable devices, such as smartwatches (illustrated in Fig. 1), showcasing the specific application focus of the tailored metasurface technology. The FMCW radar (Infineon, BGT60TR13C) with a 60-GHz center frequency is selected, which our group successfully utilized previously for near-field probing of human tissue, particularly, glucose level detection [2], [20].

5. Could the authors comment on possible solutions to this lack of uniformity?

Authors' Reply:

As addressed in responses to questions 1 and 4, non-uniformity is not a significant concern in the proposed application. This is attributed to the small dimensions of the metasurface array in relation to the human body skin across all areas. Moreover, in the context of real-time glucose monitoring with a wearable device, the device tightly encases the wrist, establishing strong contact with the human skin and rendering potential non-uniformity negligible.

However, in scenarios where uniformity is crucial for employing the presented metasurface-enhanced radar system in other biomedical applications, there exist solutions to mitigate the impact of non-uniformity.

- 1) Achieving a reduction in the metasurface array size while preserving its resolution enhancement capability offers a solution to address non-uniformity concerns. The current array, detailed in this paper, employs 7×7 unit cells, each measuring 1.1 mm, essential for providing phase compensation in the radar antenna's near-field radiation. By decreasing the effective surface of the metasurface that interfaces with the skin through a reduction in unit cell size, it is possible to maintain consistent results in a smaller metasurface area. This approach diminishes the skin contact area, effectively alleviating concerns regarding non-uniformity.
- 2) Utilizing a flexible substrate in the design of the planar metasurface, allowing it to conform to the contours of the body skin, presents an effective strategy for minimizing concerns related to non-uniformity.
- 3) Employing machine learning algorithms to account for skin variability in the received power of the radar, coupled with signal filtering techniques, offers a robust solution to extract valuable information from radar signals while mitigating interference arising from skin non-uniformity.
- 4) Creating a wideband metasurface for the radar system, capable of operating across a range of frequencies, offers advantages. Diverse frequencies can penetrate the skin to different depths and may interact distinctively with skin features, potentially yielding a more comprehensive perspective.

6. Together with the metasurface, the radar system is presented with a detection algorithm. The algorithm is pretty simplistic. How does the system effectively performs when measuring one or multiple targets inside the skin? How well can they be located with and without the addition of the metasurface?

Authors' Reply:

The study exclusively employed a singular receiver as there existed only one target within the near field of the system. In circumstances demanding differentiation among multiple targets, the utilization of multiple receivers would have been necessary. Throughout simulations and practical experiments, it was observed

that the RX1 and RX2 receivers were impacted by the leakage originating from the transmitter. Consequently, the RX3 receiver was selected for the experimental validations.

Furthermore, it is essential to highlight that the designed metasurface possesses the capability to concentrate power into a focal point in the near field of the radar antenna, facilitating high sensitivity for a single target. The prospect of measuring multiple targets is an interesting area for potential future exploration.

Regarding the performance of the system for sensing a single target, referring to the sections “Near-field analysis of the radar system integrated with and without metasurface” and “Fabrication process and measurement results” in the manuscript, a comprehensive discussion is presented by reporting the simulations and measurements results for the radar system sensing in both cases: integrated with and without the metasurface.

As reported in the manuscript:

Simulations Analysis: The proposed structure is simulated using a full wave electromagnetic simulator, HFSS, and the results of near electric field magnitude and focused power density of the radar system in the presence or absence of the proposed transmitarray metasurface is obtained inside the pure water modelled as a cylindrical slab. For this model, the maximum peak of the power density occurs at a 2 mm distance above the metasurface, 1 mm above the beaker bottom, inside the pure water model. As shown in Fig. 9, the designed metasurface provides a significant enhancement on the near electric field inside the water which is 8.8 times higher than the radar antenna without metasurface. It is expected that the near-electric-field enhancement provides a very directive near-field power density with high intensity. Fig. 10a and 10b show the 2D contour plots of the power density radiated by the TX radar antenna in the presence and absence of the near-field-focused planar metasurface in a rectangular region, lying on the plane in parallel to the array with a 2 mm distance. It is apparent that using the near field-focused metasurface provides 11.5 dB (more than 14 times) improvement in the radiated power density across a z-constant surface above the metasurface inside the beaker-filled with pure water model at 60 GHz. The S-parameter analysis of the reflection and transmission coefficients, $S(TX-TX)$ and $S(TX-RX3)$, for the radar system with and without metasurface, are also investigated. Fig. 11 shows that utilizing metasurface and considering the model of the beaker filled with pure water provides 11.5 dB power reflected enhancement from the water medium to one of the radar receivers, RX3, leading to significant enhancement of the radar SNR.

Measurements: The measurement results showing the power transferred into the water medium and picked up by the probe are presented and compared in Fig. 14a. As observed, using the designed transmitarray metasurface enhances the power received by the probe at the operational radar frequency range, significantly. Looking specifically at the designed frequency of 60 GHz shows an improvement of 11 dB in the measured near-field power, which is in good agreement with the simulation results representing the power density improvement of 11.5 dB in Fig. 10. The next step is the radar SNR investigation by measuring the reflected power from the water to one of the radar receiver antennas (RX3). The algorithm 1 is used to measure the received power in a period of time in the cases of the radar system loaded by a beaker filled with water in the presence or absence of the transmitarray metasurface. To illustrate near-field sensing improvement with higher SNR, different concentration levels of sugar are added to water and the results are analyzed and compared in Fig. 14b. As shown, using the near-field-focused metasurface enhances the power reflection from the pure water by 11.3 dB and improves the radar SNR around 13.4-times, which is in good agreement with the simulation results presented in Fig. 11. Furthermore, in this

experimental study, the effect of varying the dissolved amount of sugar in water in the presence of the metasurface is explored. 200 experiments for each scenario were carried out to provide high repeatability. Although power reflection is reduced by increasing sugar content in both cases, the metasurface clearly improves the received power level by the radar as shown in Fig. 14b. The received power in the presence of the metasurface for different sugar concentration levels has a resolution of around 0.5 dB per each 15 mg/ml concentration, validating that the metasurface enhances the overall sensing functionality.

7. There are several typos in the text and the symbols used for the equations (e.g., different symbols to indicate the phase shift ϕ).

Authors' Reply:

Thank you for your feedback and for highlighting the concerns regarding typos in the text. We have thoroughly reviewed and addressed these issues. The text has undergone a comprehensive review and meticulous editing to rectify any typos and inconsistencies in symbol representation, particularly ensuring uniformity in indicating the phase shift within the equations.

Comments from Reviewer 2:

The paper is well-written and effectively communicates the research findings, contributing significantly to the existing body of knowledge in the field.

Dear Reviewer,

We express our sincere appreciation for your considerate assessment of our manuscript. Thank you for your valuable feedback and encouraging remarks regarding our paper. Your insights and comments are instrumental in our endeavor to make meaningful contributions to the field and enhance the quality and clarity of our manuscript. We acknowledge your valuable suggestions for improvement and the identification of areas that require correction. We have meticulously proofread the manuscript and addressed all the issues you have highlighted.

Comments to be addressed:

1. Has the suggested unit cell been previously introduced? What distinguishes it in terms of novelty and advantages when compared to other existing studies?

Authors' Reply:

The suggested unit cell introduced in our research draws inspiration from and shares conceptual ties with the well-known crossed dipole antenna, which is a recognized concept in literature. Crossed dipole antennas have been widely developed for current and future wireless communication systems. They can generate omnidirectional, dual-polarized, and circularly polarized radiation. Moreover, by incorporating a variety of primary radiation elements, they are suitable for single-band, multiband, and wideband operations. Having these features, we can take advantage of this design by considering its microstrip model. The microstrip crossed dipole unit cell has been previously used in the design of the frequency selective surfaces [R1], reflectarray antenna [R2], and transmitarray antenna [R3] to provide high efficiency

and dual-band features into the space-fed array antennas. In addition, in applications required in the far field, the crossed dipole antennas can offer more symmetric and well-defined radiation patterns compared to some other designs. This can aid in achieving more consistent and predictable radiation characteristics.

In this paper, the crossed dipole structure was employed for the unit cell, due to the reasons detailed in the following explanation:

- 1) **Practical Design:** The proposed unit cell stands out from those explored in prior studies, introducing a novel perspective to the field. In the metasurface design for biomedical applications, a notable challenge in unit cell design arises from reflections at the air-skin interface, leading to significant mismatching. This distinctive characteristic sets the design process apart from typical metasurface designs documented in the literature. Achieving an effective design to enhance the electric field intensity penetrating the body skin requires careful consideration of the actual skin model in unit cell design and related simulations. This ensures proper impedance matching when the unit cell operates in contact with the skin medium rather than in free space. Comparing wave impedances inside the human skin model and wave impedance in the metasurface indicates that the design parameters of the metasurface are optimized such that providing a capacitive behavior within the desired frequency band and, in particular, a value close to the skin model impedance but opposite in sign at 60 GHz. Therefore, the designed metasurface unit cell is able to counteract the inductive behavior of the human skin model providing high impedance matching between the radar antenna propagation and the skin model leading to a higher transferred power into the skin medium, and higher reflected power received by the RX antenna.
- 2) **Size Reduction:** For the unit cell with body skin model providing the advantage of the smaller unit cell dimensions leading to a low-profile array while reducing phase error losses. In the design of the unit cell working in free space [R1][R2][R3], typical values of the length and width are considered as 0.5λ (λ is the free-space wavelength) with a 1.25 mm substrate thickness, while the dimensions of the designed unit cell with human body skin model reduce to 0.22λ with a 0.8 mm substrate thickness.

- 3) **Providing one focused spot with high resolution:** The other advantage of unicell size reduction is to ensure that $W_f < \lambda$ avoids more than one spot region appearing in the near-focused region, as a similar concept to the grating lobes occurs in the far-field radiation pattern of any uniformly spaced array [R2][R3] enabling a metasurface with a high power focused focal point for better biomedical sensing.

- 4) Higher efficiency by resilience to Incidence Angle Variations: Crossed dipole unit cells can maintain relatively stable performance over a range of incident angles. This characteristic makes them suitable for applications where the incoming signal angle may vary, such as in near-field biomedical sensing. Taking advantage of the low-profile unit cell with 0.22λ dimension, when the unit cells are illuminated at various locations, the oblique incidents have a very low impact on the magnitude and phase of the unit cell. For the proposed metasurface placed at $\lambda/2$ distance, the maximum incidence angles up to 70° required at the edge elements cause the phase deviation lower than 6° , while the magnitude makes only 0.2 dB variations. Therefore, the phase error loss caused by the oblique incident is negligible, which simplifies the design process while guaranteeing nondestructive effects on the phase difference compensation.
- 5) Multi-polarization design: Crossed dipole structure inherently produces dual linearly polarized (horizontal and vertical) as well as circularly polarized radiation due to its symmetrical structure. In the near-field region of an antenna, the emitted electromagnetic fields exhibit complex polarization characteristics, often displaying polarization in different directions. This phenomenon arises due to the intricate interaction of electromagnetic waves in the near-field zone, where the electric and magnetic field components vary significantly with proximity to the antenna. In such a case, it is essential to have a metasurface design that supports and accommodates this diverse range of polarization states, ensuring adaptability to the variable polarization conditions encountered in the near-field zone. Understanding these diverse polarization characteristics in the near field is fundamental for biomedical sensing applications where the specific polarization states play a significant role in determining system performance and functionality. Using a crossed dipole structure allows passing near-field waves with any polarizations leading to higher electric field intensity into the skin medium and as a result, increasing power transferring and higher sensing.
- 6) Ease of Fabrication and Integration: Crossed dipole unit cells are often simpler to manufacture and integrate into the metasurface structure, making them a practical choice for large-scale production and deployment.
- 7) Reducing undesired reflections from the metal layer: Given that the design is based on the near-field operation and considering a microstrip metasurface design, it is anticipated that undesired reflections may occur when the metallic layer covers a significant portion of the metasurface. In such instances, it is suitable to consider a unit cell design that minimizes the metallic coverage on the surface. Selecting the unit cell design with the crossed dipole structure reduces the metallic coverage, mitigating undesirable reflections, which could otherwise interfere with the intended performance of the metasurface structure.

[R1] Zhuang, W., Fan, Z., Ding, D. Z., An, Y. Fast analysis and design of frequency selective surface using the GMRESR-FFT method. *Progress In Electromagnetics Research B*, 12, 63-80 (2019).

[R2] Liu, X., Ge, Y., Chen, X., & Chen, L. Design of folded reflectarray antennas using pancharatnam-berry phase reflectors. *IEEE Access*, 6, 28818-28824 (2018).

[R3] Bagheri, M. O., Hassani, H. R., Rahmati, B. Dual-band, dual-polarised metallic slot transmitarray antenna. *IET Microwaves, Antennas and Propagation*, 11, 3, 402-409 (2017).

2. It's crucial to delve into the design intricacies and features of the 4-element radar antenna.

Authors' Reply:

While we deeply appreciate the significance of exploring the design intricacies and features of such radar antennas, our current focus in this paper has not involved the direct design and development of a 4-element radar antenna.

This paper introduces a low-profile planar metasurface which is specifically designed with impedance matching capabilities, seamlessly integrated with radar transmitter and receiver antennas, facilitating direct contact with a human body simplified model to enable substantial advancements in near-field sensing performance for biomedical applications. The key aspects contributing to the metasurface's remarkable radar near-field sensing were the enhanced power density absorption from the radar antenna transmitter into a controlled medium alongside the elevated received power level by the radar antenna receiver, leading to a higher system signal-to-noise ratio. Specifically, inside the simplified liquid container, the use of the metasurface led to an improvement of more than 11 dB in the near-field Poynting power density. Moreover, through radar signal processing, the analysis revealed an additional improvement of over 11 dB in radar signal-to-noise ratio, thereby enhancing the sensor sensing abilities.

While the radar used in this paper, Infineon BGT60TR13C chipset includes four on-chip antennas, one of which radiates as a transmitter and the others of which serve as receivers, the study exclusively employed a singular receiver (RX3) as there existed only one target within the near field of the system. In circumstances demanding differentiation among multiple targets, the utilization of multiple receivers would have been necessary. Throughout simulations and practical experiments, it was observed that the two other receivers were less affected by the leakage originating from the transmitter. Consequently, the third receiver was selected for the experimental validations.

3. Furthermore, there needs to be an in-depth discussion about the distance between the source antenna and the metasurface layer.

Authors' Reply:

The manuscript includes a discussion on the distance between the radar antenna as a source and the metasurface layer, as follows:

“To achieve a concentrated near-field power, the radar range which is the distance between the feeding antenna and the metasurface in contact with the skin model is crucial. If an airgap distance of $\lambda/2$ (λ in free space) is considered, in accordance with the Fabry-Perot Cavity (FPC) theory, the proposed metasurface is employed in the configuration of an FPC which reduces backscattering and increases the peak radiation of the underlying source antenna leading to enhance transferred power density into the skin [50]. As a result, placing the metasurface and skin model at the range of $\lambda/2$ at 60 GHz, 2.5 mm air gap above the radar antenna, provides the highest power concentration into human skin.”

Discussion about Fabry-Perot cavity (FPC) theory:

As shown in the figure below, a conventional FPC resonator antenna is formed by placing a structure as a partially reflective surface (PRS) at a proper distance from the ground plane, which creates an air-filled

cavity between the PRS and the ground plane and fed by a small antenna or an array [R1]. According to the ray theory, the main characteristics of the antenna radiation are determined by the property of the PRS and the distance to the source. Part of the electromagnetic waves radiated by the antenna passes through the coating directly, and the other part is reflected once or several times before passing through the coating. By satisfying the following equation, electromagnetic waves become superimposed in phase, leading to the enhancement of power on the opposite side [R2].

$$h = \frac{c}{4\pi f} + (\varphi_{PRS} + \varphi_{GND} - 2N\pi) , N = 0,1,2,\dots$$

where h is the depth of the cavity, f is the resonant frequency of the antenna, c is the velocity of light, N is the order of the resonance mode, φ_{PRS} is the reflection phase of the cladding structure, and φ_{GND} is the reflection phase of the ground plane. Normally, φ_{GND} is π due to its almost full reflection, and φ_{PRS} can be derived from as follows:

$$\varphi_{PRS} = \frac{4\pi h}{c} f + (2N - 1)\pi , N = 0,1,2,\dots$$

As indicated, the reflection phase of the cladding structure is positively correlated to the frequency with a reflection slope of $4\pi h/c$. Utilizing the previously presented formulas, one can calculate that by selecting the zero-order mode ($N = 0$) and setting the air gap distance to half-wavelength ($h = \lambda/2$), the resulting phase φ_{PRS} is equal to π .

Hence, by applying the discussed Fabry-Perot theory, when a source is emitted, all waves undergo a 180-degree phase shift at the PRS surface. Consequently, we can categorize the waves into two groups: 1) waves passing through the PRS without reflection, and 2) waves that are partially reflected from the PRS. In the second category, when the reflected waves reach the source ground, they once again experience a 180-degree phase shift. This implies that the reflected waves from the PRS accumulate a total phase shift of 360 degrees, aligning them in phase with the initial waves radiating from the source. Consequently, this structure manages to maintain the coherence of reflected waves, reinforcing them in phase with the primary radiation, ultimately amplifying the overall power on the opposite side of the PRS.

Therefore, in this paper placing the metasurface and skin model at the range of $\lambda/2$ at 60 GHz, adding a 2.5 mm air gap above the radar antenna provides the highest power concentration into the medium, here represented by human skin. This assertion is supported by the simulations presented below. Here, the result of the transferred power into the human skin medium, when the metasurface is placed at the distance of 0.5λ , is compared to the results of the transferred power into the skin medium when the metasurface is placed at the distance of 0.6λ and 0.4λ . The results show that increasing the distance by

0.1 λ and decreasing the distance by 0.1 λ resulted in a reduction of the maximum focused power into the skin by 1.1 dB and 1.7 dB, respectively.

$h = 0.5 \lambda$ (Optimal distance)

$h = 0.4 \lambda$

$h = 0.6 \lambda$

[R1] Umair, H., Latef, T. B. A., Yamada, Y., Mahadi, W. N., Othman, M., Kamardin, K., Najam, A. I. Tilted beam Fabry–Perot antenna with enhanced gain and broadband low backscattering. *Electronics*, 10, 3, 267 (2021).

[R2] Sheng, X., Lu, X., Liu, N., Liu, Y. Design of Broadband High-Gain Fabry–Perot Antenna Using Frequency-Selective Surface. *Sensors*, 22, 24, 9698 (2022).

4. Additionally, it is imperative to include a comparison table with informative descriptions to thoroughly discuss and emphasize the novelty and advantages of the proposed method and the designed metasurface.

Authors’ Reply:

Thank you for your valuable feedback. We have incorporated a dedicated table in the manuscript, Table 2, at the end of the section: “Near-field transmitted power density and radar SNR measurement”. This table provides informative descriptions that discuss and emphasize the novelty and advantages of the proposed method using the designed metasurface.

Table 2: Novel features and advantages of utilizing the transmissive metasurface method for biomedical sensing.

Features	Descriptions
Near-Field Theory Considerations	The metasurface is designed with a methodology tailored for near-field radiation theory, essential for biomedical sensing, in contrast to prior works focusing on the usual far-field antenna radiation.
Compactness	In near field focusing, current antenna designs face integration challenges with low-profile radar due to their bulky structures. This paper proposes a low-profile, compact, and planar metasurface for seamless integration with a radar system.
Millimeter-wave Operation	The mm-wave frequency range is selected for its high resolution and precision, biocompatibility, and reduced interference making it suitable for applications in wearable devices like smartwatches.
Direct Human Skin Contact	Near-field studies typically focus on antenna radiation in free space, but the human body causes detuning and impedance mismatching, degrading performance. Existing methods are unsuitable for direct-contact biomedical applications. The designed metasurface allows skin contact, enabling precise targeting without interference.
Impedance Matching Layer	The metasurface is carefully designed based on impedance matching network theory to achieve highly effective impedance matching between free space and the human skin. This design tackles a significant challenge in the literature by minimizing reflections arising from air-skin interference.
Intensify Radiated Near Electric Field	The metasurface, with phase-synthesized unicells, significantly boosts the near-field electric field from the source antenna, achieving an 8.8-fold improvement within the proposed skin phantom medium compared to the radar antenna without the metasurface.
Intensify Near-field Transmitted Power	The metasurface enhances absorbed power in human skin at 60 GHz without affecting source antenna impedance matching. This near-field-focused design yields an 11 dB improvement in radiated power density above the metasurface within the proposed skin phantom medium.
Intensify Near-field Reflected Power	Analyzing the radar's reflection coefficient shows the metasurface amplifying reflected power from human skin by 11.3 dB. This enhancement leads to a notable 13.4-times improvement in radar SNR, enabling effective information transmission and high-performance near-field sensing in biomedical applications.
Glucose Sensing Enhancement	In glucose monitoring, metasurface-enhanced power measurements show a resolution of around 0.5 dB per 15 mg/ml sugar concentration, maintaining a high SNR and confirming enhanced functionality in glucose sensing.

5. Furthermore, the references should be updated to incorporate more recently published papers, preferably from 2023, for a comprehensive and current overview of the field.

Authors' Reply:

We appreciate your suggestion for a more comprehensive overview of the field. In response to your comment, we have updated the references in the manuscript, incorporating more recently published papers, including those from 2022 and 2023. This adjustment aims to ensure that the references provide an up-to-date and relevant foundation for our work.

2022 References:

[26]	Bagheri, M. O., and Shaker, G. Near-Field Enhancement of Microstrip Antenna Using Engineered Superstrate for Biomedical Applications. In 2022 International Symposium on Antennas and Propag. and USNC-URSI Radio Science Meeting (AP-S/URSI) , 1038-1039 (IEEE, 2022).
[37]	Bagheri, M. O., Hassani, H. R., and Sebak, A. R. Stable phase-centre horn antenna using 3D printed dielectric rod for aperture efficiency improvement of space-fed antennas. IET Microwaves, Antennas and Propag. , 16, 14, 888-897 (2022).
[41]	Li, H., et al. High-Resolution Near-Field Imaging and Far-Field Sensing Using a Transmissive Programmable Metasurface. Advanced Materials Tech. 7, 5, 2101067 (2022).
[44]	Alibakhshikenari, M., et al. A comprehensive survey on antennas on-chip based on metamaterial, metasurface, and substrate integrated waveguide principles for millimeter-waves and terahertz integrated circuits and systems. IEEE Access. 10, 3668-3692 (2022).
[48]	Xiao, L., Xie, Y., Gao, S., Li, J., and Wu, P. Generalized Radar Range Equation Applied to the Whole Field Region. Sensors. 22, 12, 4608 (2022).

2023 References:

[7]	Reggad, H., Jiang, X., Wu, X., Amirtharajah, R., Matthews, D., Liu, X. A Single-Chip Single-Antenna Radar for Remote Vital Sign Monitoring. IEEE Trans. on Microwave Theory and Tech. (2023).
[9]	Mercuri, M., Soh, P. J., Mehrjousesht, P., Crupi, F., and Schreurs, D. Biomedical Radar System for Real-Time Contactless Fall Detection and Indoor Localization. IEEE Journal of Electromag., RF and Microwaves in Medicine and Biology (2023).
[10]	Xiao, Z., Yang, C., Li, Y., Xing, Y., Ma, C., Zhang, Y., Liu, C. Human Eye Activity Monitoring Using Continuous Wave Doppler Radar: A Feasibility Study. IEEE Trans. on Biomedical Circuits and Systems (2023).
[11]	Ullah, R., Saied, I., Arslan, T. Multistatic radar-based imaging in layered and dispersive media for biomedical applications. Biomedical Signal Processing and Control , 82, 104568 (2023).
[13]	Brizi, D., Conte, M., and Monorchio, A. A Performance-Enhanced Antenna for Microwave Biomedical Applications by Using Metasurfaces. IEEE trans. Antennas and Propag. 71, 3314-3323 (2023).
[20]	Abu-Sardanah, S., Gharamohammadi, A., Ramahi, O. M., and Shaker, G. A Wearable mm-Wave Radar Platform for Cardiorespiratory Monitoring. IEEE Sensors Let. , 7, 6, 1-4 (2023).
[21]	Kandwal, A., Liu, L. W., Deen, J., Jasrotia, R., Kanaujia, B. K., Nie, Z. A. Electromagnetic Wave Sensors for Non-Invasive Blood Glucose Monitoring: Review and Recent Developments. IEEE Trans. on Instrumentation and Meas. (2023).
[22]	Kirubakaran, S. J., Bennet, M. A., Shanker, N. R. Non-Invasive antenna sensor based continuous glucose monitoring using pancreas dielectric radiation signal energy levels and machine learning algorithms. Biomedical Signal Processing and Control. , 85, 105072 (2023).
[58]	Bagheri, M. O., Ramahi, O. M., and Shaker, G. Near-Field Sensing Improvement of a Radar Antenna-On-Chip Using a Transmissive Superstrate. In 2023 URSI International Symposium on Electromag Theory. (IEEE, 2023).

Comments from Reviewer 3:

In this work the metasurface is integrated with a mm-wave radar chipset from Infineon and a human skin model to improve the radar performance. Authors demonstrate that their metasurface study enhances the power density absorption and reflection, as well as the signal-to-noise ratio, of the radar system when in contact with a human body model.

The paper is written well, however it lacks biomedical related content. It mainly focuses on study of antenna design and the impact of metasurface study.

Dear Reviewer,

We sincerely appreciate your thoughtful evaluation of our manuscript and are grateful for the valuable feedback and positive remarks you provided. Your insights play a crucial role in our ongoing efforts to make substantive contributions to the field and improve the overall quality and clarity of our paper. We recognize the importance of your suggestions for refinement and the identification of areas requiring correction. With careful attention, we have thoroughly proofread the manuscript and addressed all the issues you highlighted.

Specifically, in response to your observation about the lack of biomedical content, we have significantly expanded the manuscript to thoroughly explore biomedical applications related to antenna design and metasurface study. Detailed discussions on the potential implications of our findings within the context of biomedical technologies, along with specific applications, have been incorporated in this response letter as well as the manuscript.

The methodology presented, metasurface-enhanced radar near-field sensing, demonstrates broad applicability in diverse biomedical sensing contexts including glucose monitoring [2], [15], [16], skin cancer [18], as well as heart and on-body radar cardiorespiratory monitoring [20]. Characterized by elevated blood glucose levels, diabetes is a prevalent and chronic condition that underscores the crucial need for early detection and diagnosis. Despite the availability of invasive techniques, the increasing focus on non-invasive glucose measurement, offering additional benefits without causing inconvenience to the human body, drives ongoing research and the exploration of new possibilities [21].

Our research group has been actively engaged in diverse applications related to biosensing applications, specifically non-invasive glucose monitoring. In [2], a novel design of a portable planar microwave sensor operating within the ISM band 2.4–2.5 GHz is introduced, enabling rapid, accurate, and non-invasive blood glucose level monitoring. [15] introduces an integrated millimeter-wave radar system to detect different glucose concentration levels in artificial blood samples. This study aims to affirm the suitability of mm-wave radars for non-invasively monitoring diabetes patients' glucose levels, leveraging signal processing approaches to identify various glucose concentrations and correlate them with the reflected mm-wave readings. In the work detailed in [16], a novel approach to monitoring glucose levels is presented, utilizing a robust low-power millimeter-wave radar system to differentiate between blood samples with varying glucose concentrations. [22] proposes an alternative glucose monitoring approach using an antenna sensor operating at 4.2 GHz positioned on the pancreas to capture dielectric radiation signals. [23] details the development of a microwave biosensor for real-time non-invasive glucose monitoring operating in the range of 1–6 GHz.

As explored in the literature, monitoring blood glucose levels non-invasively addressing high sensitivity and high resolution has posed a considerable challenge. This paper specifically emphasizes the potential of realizing high-sensitivity mm-wave radar for real-time disease monitoring, with glucose sensing for diabetic patients as an example. The vision of this work is it will enable glucose monitoring capabilities into wearable devices as shown in Fig. 1, such as smartwatches, emphasizing the specific application context for which this metasurface technology is tailored. The FMCW radar (Infineon, BGT60TR13C) with a 60-GHz center frequency is selected, which our group successfully utilized previously for near-field probing of human tissue, particularly, glucose level detection [2], [20], [58]. As presented in the manuscript, Fig. 14b shows the analysis of the power received by the radar RX antenna from the proposed phantom (beaker filled with pure water) in the presence or absence of the designed transmissive metasurface for different concentration levels of sugar.

The introduction section of the manuscript has been updated (highlighted in red) by adding Fig. 1 and explaining the application related to this work.

Figure 1 (a) A wearable device equipped with radar technology positioned near a human hand; (b) Incorporating a metasurface structure between the radar antennas and the hand to enhance the near-field energy coupling and Signal-to-Noise Ratio (SNR).

Figure 14(b) The investigation of the power reflected from the beaker-filled with pure water received by RX3 radar antenna over a period in the presence or absence of the designed transmissive metasurface for different concentration levels of sugar.

Some specific comments:

1. In Figure 3, the simulations indicate that the ports are placed on both ends of the model, like a transmittance model. Whereas in the measured scenario based on Fig. 7, the receiver antenna RX3 and transmitting antenna TX are on the same end. Would the results be the same as that of the earlier simulation?

Authors' Reply:

This paper introduces a planar metasurface which is specifically designed with impedance matching capabilities, seamlessly integrated with radar transmitter and receiver antennas, facilitating direct contact with a human body simplified model to enable substantial advancements in near-field sensing performance for biomedical applications. The key aspects contributing to the metasurface's remarkable radar near-field sensing are the enhanced power density absorption from the radar antenna transmitter into a controlled medium as well as the elevated received power level by the radar antenna receiver, leading to a higher system signal-to-noise ratio. To provide a high absorbed power inside the human skin model, the metasurface needs to be designed according to the transmitarray theory. The design process and performance of the metasurface are examined, taking into account the design of the unit cell and the analysis of the phase-synthesized array.

The design of the transmissive unit cell is presented in Fig. 4 (previously Fig. 3). The Floquet port simulation as a computational technique used in electromagnetic simulations, particularly for periodic structures, is utilized to analyze the unit cell performance in both transmission and reflection modes. In the unit cell simulation presented in Fig. 4, two ports at both ends are located providing us with the S-parameter matrix for a 2×2 network which is: S_{11} , S_{12} , S_{21} , and S_{22} .

In this specific case, the analysis of the unit cell's transmission ability involves considering the parameters S_{11} and S_{21} . This signifies examining how much power is reflected to port 1 and how much is received by port 2 when port 1 radiates. The designed unit cell, connected to the human body skin model, is positioned between these two ports for evaluation. Compared to Fig. 8 (previously Fig. 7), this analysis aligns precisely with the evaluation of absorbed power into the human skin when the radar TX antenna emits radiation. Ensuring a suitable reflection coefficient (below -10 dB) and transmission coefficient (close to 0 dB) in the unit cell design (Fig. 4) guarantees high power transmission and concentration into the body skin. The reflection (Fig. 5a) and transmission (Fig. 7a) coefficients of the designed unit cell demonstrate strong performance in the presence of the human skin medium. Consequently, the incorporation of the proposed unit cell in the transmissive metasurface array design results in an improvement of over 11 dB in the near-field absorbed power density, particularly within the simplified beaker filled with pure water.

As the power within the human skin increases to enhance the radar's sensing capability using the metasurface, the radar's receiver side (RX3) also requires higher power reception. In this scenario, the metasurface is designed to function as a matching network between the radar antennas and the body skin, aiming to reduce power reflection from the metasurface and skin interface and increase power reflection from inside the skin. Consequently, the metasurface facilitates the transmission of power from inside the skin back to the radar receiver antenna. Through radar signal processing, the analysis reveals an additional improvement of over 11 dB in received power and the radar signal-to-noise ratio as a result.

Therefore, in both aspects of the metasurface design, the unit cell needs to provide good performance in the transmission mode, initially, efficiently transferring power from the radar TX antenna to the skin, and subsequently, effectively transmitting reflected power from the skin to the radar RX antenna. This is essential for ensuring high sensing capabilities in biomedical applications.

2. Could you elaborate on the significance of crossed dipole unit cell? How do you compare them with literature? Are there any additional advantages?

Authors' Reply:

The suggested unit cell introduced in our research draws inspiration from and shares conceptual ties with the well-known crossed dipole antenna, which is a recognized concept in literature. Crossed dipole antennas have been widely developed for current and future wireless communication systems. They can generate omnidirectional, dual-polarized, and circularly polarized radiation. Moreover, by incorporating a variety of primary radiation elements, they are suitable for single-band, multiband, and wideband operations. Having these features, we can take advantage of this design by considering its microstrip model. The microstrip crossed dipole unit cell has been previously used in the design of the frequency selective surfaces [R1], reflectarray antenna [R2], and transmitarray antenna [R3] to provide high efficiency and dual-band features into the space-fed array antennas. In addition, in applications required in the far field, the crossed dipole antennas can offer more symmetric and well-defined radiation patterns compared to some other designs. This can aid in achieving more consistent and predictable radiation characteristics.

In this paper, the crossed dipole structure was employed for the unit cell for nearfield sensing (which is not its typical use in literature), due to the reasons detailed in the following explanation:

- 1) **Practical Design:** The proposed unit cell stands out from those explored in prior studies, introducing a novel perspective to the field. In the metasurface design for biomedical applications, a notable challenge in unit cell design arises from reflections at the air-skin interface, leading to significant mismatching. This distinctive characteristic sets the design process apart from typical metasurface designs documented in the literature. Achieving an effective design to enhance the electric field intensity penetrating the body skin requires careful consideration of the actual skin model in unit cell design and related simulations. This ensures proper impedance matching when the unit cell operates in contact with the skin medium rather than in free space. Comparing wave impedances inside the human skin model and wave impedance in the metasurface indicates that the design parameters of the metasurface are optimized such that providing a capacitive behavior within the desired frequency band and, in particular, a value close to the skin model impedance but opposite in sign at 60 GHz. Therefore, the designed metasurface unit cell is able to counteract the inductive behavior of the human skin model providing high impedance matching between the radar antenna propagation and the skin model leading to a higher transferred power into the skin medium, and higher reflected power received by the RX antenna.
- 2) **Size Reduction:** For the unit cell with body skin model providing the advantage of the smaller unit cell dimensions leading to a low-profile array while reducing phase error losses. In the design of the unit cell working in free space [R1][R2]R3], typical values of the length and width are considered as 0.5λ (λ is the free-space wavelength) with a 1.25 mm substrate thickness, while the dimensions of the designed unit cell with human body skin model reduce to 0.22λ with a 0.8 mm substrate thickness.

- 3) Providing one focused spot with high resolution: The other advantage of unicell size reduction is to ensure that $W_f < \lambda$ avoids more than one spot region appearing in the near-focused region, as a similar concept to the grating lobes occurs in the far-field radiation pattern of any uniformly spaced array [R2][R3] enabling a metasurface with a high power focused focal point for better biomedical sensing.
- 4) Higher efficiency by resilience to Incidence Angle Variations: Crossed dipole unit cells can maintain relatively stable performance over a range of incident angles. This characteristic makes them suitable for applications where the incoming signal angle may vary, such as in near-field biomedical sensing. Taking advantage of the low-profile unit cell with 0.22λ dimension, when the unit cells are illuminated at various locations, the oblique incidents have a very low impact on the magnitude and phase of the unit cell. For the proposed metasurface placed at $\lambda/2$ distance, the maximum incidence angles up to 70° required at the edge elements cause the phase deviation lower than 6° , while the magnitude makes only 0.2 dB variations. Therefore, the phase error loss caused by the oblique incident is negligible, which simplifies the design process while guaranteeing nondestructive effects on the phase difference compensation.
- 5) Multi-polarization design: Crossed dipole structure inherently produces dual linearly polarized (horizontal and vertical) as well as circularly polarized radiation due to its symmetrical structure. In the near-field region of an antenna, the emitted electromagnetic fields exhibit complex polarization characteristics, often displaying polarization in different directions. This phenomenon arises due to the intricate interaction of electromagnetic waves in the near-field zone, where the electric and magnetic field components vary significantly with proximity to the antenna. In such a case, it is essential to have a metasurface design that supports and accommodates this diverse range of polarization states, ensuring adaptability to the variable polarization conditions encountered in the near-field zone. Understanding these diverse polarization characteristics in the near field is fundamental for biomedical sensing applications where the specific polarization states play a significant role in determining system performance and functionality. Using a crossed dipole structure allows passing near-field waves with any polarizations leading to higher electric field intensity into the skin medium and as a result, increasing power transferring and higher sensing.
- 6) Ease of Fabrication and Integration: Crossed dipole unit cells are often simpler to manufacture and integrate into the metasurface structure, making them a practical choice for large-scale production and deployment.
- 7) Reducing undesired reflections from the metal layer: Given that the design is based on the near-field operation and considering a microstrip metasurface design, it is anticipated that undesired reflections may occur when the metallic layer covers a significant portion of the metasurface. In such instances, it is suitable to consider a unit cell design that minimizes the metallic coverage on the surface. Selecting the unit cell design with the crossed dipole structure reduces the metallic coverage, mitigating undesirable reflections, which could otherwise interfere with the intended performance of the metasurface structure.

[R1] Zhuang, W., Fan, Z., Ding, D. Z., An, Y. Fast analysis and design of frequency selective surface using the GMRESR-FFT method. Progress In Electromagnetics Research B, 12, 63-80 (2019).

[R2] Liu, X., Ge, Y., Chen, X., & Chen, L. Design of folded reflectarray antennas using pancharatnam-berry phase reflectors. IEEE Access, 6, 28818-28824 (2018).

[R3] Bagheri, M. O., Hassani, H. R., Rahmati, B. Dual-band, dual-polarised metallic slot transmitarray antenna. *IET Microwaves, Antennas and Propagation*, 11, 3, 402-409 (2017).

3. Are there any effects from antenna radiation pattern, directivity, and bandwidth on the near-field sensing performance?

Authors' Reply:

The usual antenna radiation parameters such as pattern, directivity and bandwidth are primarily applicable in the far-field region of an antenna. The near field, which is close to the antenna aperture, has different electromagnetic characteristics that make these parameters less meaningful or not applicable. Here are the key reasons:

- 1) In the far-field, the electromagnetic fields are characterized by both electric and magnetic field components that are in phase and perpendicular to each other. These fields propagate as electromagnetic waves. In the near field, however, the fields are more complex. There are reactive (non-radiating) fields and radiating fields. The reactive near-field is dominated by the electric and magnetic fields, while the radiating near-field starts to exhibit the characteristics of far-field radiation.
- 2) The electromagnetic waves in the far field propagate independently of the antenna structure. They follow the inverse square law, where the power density decreases with the square of the distance from the antenna. The near-field is influenced by the geometry and characteristics of the antenna. The fields in this region do not necessarily follow the inverse square law.
- 3) In the near field, the electromagnetic fields are complex and vary spatially. The fields have both reactive (non-radiating) and radiating components. The near field is characterized by strong electric and magnetic fields that are not yet fully developed into the well-defined electromagnetic wave patterns seen in the far field.
- 4) In the near field, the reactive near-field dominates, and energy is stored in the electric and magnetic fields rather than being radiated away as electromagnetic waves. The radiation pattern, which describes the spatial distribution of radiated power in the far field, is not well-defined in the near field due to the presence of non-radiating reactive components.
- 5) The applicability of parameters like directivity and radiation pattern is closely tied to the concept that the antenna size is much smaller than the wavelength of the radiated signal in the far field. In the near field, where the distance from the antenna is comparable to the size of the antenna or even smaller, the traditional far-field assumptions do not hold.

The concept of radiation bandwidth (in contrast to impedance bandwidth) is applicable both in the near field and far field of an antenna, but its interpretation and implications may differ in these regions. In the far field, the concept of bandwidth is more commonly associated with the frequency range over which the antenna radiates efficiently. It is often expressed in terms of the frequency range where the radiation pattern, gain, and other performance parameters meet specified criteria. Therefore, in the far field, the radiation bandwidth is often associated with the ability of the antenna to radiate effectively at different frequencies, and this may be influenced by the antenna's geometry and radiation pattern characteristics. However, in the near field, the radiation bandwidth may be influenced by the antenna's ability to efficiently couple with nearby objects or devices, and impedance-matching considerations may play a crucial role.

In this paper, our emphasis lies on near-field coupling and the enhancement of Poynting power within the human body skin in direct contact with the radar system. Depending on the application, far-field radiation parameters like directivity, gain, and radiation pattern may impact antenna performance in the near-field. Nevertheless, for this particular application, we focus on the measured focal point at 2 mm above the metasurface.

Reviewers' comments:

Reviewer #1 (Remarks to the Author):

The quality of the manuscript has been improved, but some points still need to be clarified.

1) While the effect of an air gap is considered by the authors as not significantly impacting in many applications, the information contained in the answer to the reviewers comments, but absent in the paper should be included in the manuscript. Since the application of the presented technology is not yet fully defined (only some possible examples are discussed), a better overview of its limitations is also important to be known. It would be then relevant to include the graph of the transmission coefficient as a function of the air gap thickness and clearly state the maximum gap thickness that can be tolerated to allow for the power to be focus inside the body and result in a power enhancement.

2) When the authors analyse the transmission coefficient inside the radar bandwidth they only focus on the variation of the performance at the central frequency, but it would be important to add also some data in the paper about the variability of this parameter inside the operating band, that can be up to 6 dB. It would also be necessary to clarify if the considered permittivity variation is only related to the real part or also to the conductivity.

3) Are the results of fig. 11 measured or simulated? If they are measured how is the skin medium realised? If they are simulated, is the water medium also including the shell between the radar and the water slab?

4) In Fig.11 how is the 9.5 and 11 dB difference at 60 GHz justified by a permittivity variation of 0.77 between the skin and equivalent permittivity of the water+shell? The authors compute an equivalent permittivity (real part) of the water+ shell, but what about the equivalent conductivity? Do the difference between the skin medium and the water medium fall in the 10% permittivity tolerance case. Some clarifications should be added and probably a more representative phantom would be necessary to represent realistic applications of the systems.

5) It seems that the discussion about the lacking of uniformity of the metasurface included in the answer to the reviewer did not resulted in any modification in the text. This aspect is fundamental since the authors state among other application the possibility to utilise this technology for example for cancer detection. In this case a different illumination of the target(s) may lead to inaccurate results. Please discuss this in the paper.

6) I also confirm that the technical issues from Reviewer 3 have been answered by the authors. For what concerns the first comment about the possible applications of this technology, I agree with Reviewer 3. I appreciate that in the revised version the introduction is improved by providing more context, it would be important to better indicate the advantages but also the limitations of this solution in the cited applications. Also, it would be necessary to relate the choices made during the design process to the issues related to the specific application.

Reviewer #2 (Remarks to the Author):

The authors have diligently incorporated the feedback from the review process, demonstrating a thorough and thoughtful response to the comments. Consequently, the paper is now considered suitable for acceptance in its current format.

Response Document: Radar Near-Field Sensing Using Metasurface for Biomedical Applications

Manuscript ID: COMMS-ENG-23-0421A

Authors: Mohammad Omid Bagheri, Ali Gharamohammadi, Serene Abu-Sardanah, Omar M Ramahi, and George Shaker

Dear Editor and Reviewers:

We want to express our sincere gratitude for your meticulous evaluation of our manuscript (Paper No.: COMMS-ENG-23-0421A) titled "Radar Near-Field Sensing Using Metasurface for Biomedical Applications." Your insightful comments have been invaluable, and we are genuinely appreciative of your constructive suggestions. It is encouraging to learn that our paper aligns well with the publication's objectives. We have conscientiously addressed the remaining comments and suggestions from the referees, now visually highlighted in red all related edits within the revised manuscript. These modifications are intended to enhance the overall quality and contribution of the paper. Within this response letter, we have considered each of your comments, offering detailed explanations for the rationale behind our revisions. Comments and suggestions from referees are now presented in black, with our corresponding responses and modifications indicated in blue text. We believe that the paper now more effectively addresses the concerns raised during the review process and eagerly anticipate your further insights.

Comments from Reviewer 1:

The quality of the manuscript has been improved, but some points still need to be clarified.

Dear Reviewer,

We sincerely appreciate your ongoing review of our manuscript and your recognition of the implemented improvements. Enhancing the manuscript's clarity is of utmost importance to us, and we remain dedicated to addressing any lingering concerns. Your specific feedback is invaluable, and we attempted to clarify the identified issues for a more comprehensive and accessible presentation. The manuscript has undergone a meticulous proofreading process, and all the highlighted issues have been diligently addressed.

1. While the effect of an air gap is considered by the authors as not significantly impacting in many applications, the information contained in the answer to the reviewers comments, but absent in the paper should be included in the manuscript. Since the application of the presented technology is not yet fully defined (only some possible examples are discussed), a better overview of its limitations is also important to be known. It would be then relevant to include the graph of the transmission coefficient as a function of the air gap thickness and clearly state the maximum gap thickness that can be tolerated to allow for the power to be focus inside the body and result in a power enhancement.

Authors' Reply:

In response, the revised manuscript now incorporates the relevant information in a dedicated section titled 'Sensitivity analysis in near-field-focused bio-Sensing design.' This new section comprises three sub-sections, namely i) Permittivity dynamics across inter-individual differences in human skin, ii) Comparative analysis of the proposed phantom and human skin, and iii) Impact of human skin non-uniformity on planar transmissive metasurface performance. These additions aim to provide a comprehensive exploration of the topic, enhancing the overall clarity of our study.

In the third sub-section, we presented a thorough analysis of the air gap between the metasurface and human skin. As suggested, we have incorporated a graph (Fig. 12b) illustrating the transmission coefficient's dependency on the air gap thickness in the revised manuscript. Furthermore, we explicitly state the maximum tolerated air gap thickness, a crucial factor for facilitating power focus within the body and achieving power enhancement. The added sub-section in the manuscript delves into these aspects, providing valuable insights into the impact of air gap variations on our findings.

The following sub-section was added to the revised manuscript:

“The proposed metasurface is designed as a planar interface to be used with a rigid structure; however, the human body deviates from complete planarity, introducing the possibility of an air gap between the metasurface and the skin. This metasurface-enhanced radar near-field sensing method exhibits versatility across biomedical applications, including glucose monitoring, skin cancer detection, and on-body radar cardiorespiratory monitoring. The study emphasizes the development of a highly sensitive millimeter-wave radar for real-time disease diagnostics, with a specific focus on continuous blood glucose monitoring for diabetic care. The primary objective is to integrate glucose monitoring seamlessly into wearable devices, as depicted in Fig. 1, tailoring the metasurface technology for this targeted application context.

In the context of utilizing metasurface-enhanced radar for wrist-worn wearable devices, the effective area of the skin is treated as planar, assuming minimal or no gap between the metasurface and the body. This is supported by the small effective area of the metasurface ($7.7 \times 7.7 \text{ mm}^2$) and the secure fit of devices like smartwatches, ensuring practical biomedical sensing and achieving high measurement accuracy. However, it is essential to explore the performance of the metasurface under conditions where a significant gap exists between the metasurface and the human skin, particularly relevant in other biomedical contexts such as cancer detection. This consideration arises from the fact that in certain applications, there may be a notable separation between the designed metasurface and the human body, warranting an analysis of the impact on the metasurface performance under such conditions.

The simulation analysis encompasses the transmission coefficient of the metasurface unit cell and the transmitted power enhancement of the array structure, considering varying air gap distances from 0 to 1.5 mm. Illustrated in Fig. 12b is the unit cell analysis under different air gap conditions. The findings reveal that a 0.5 mm air gap induces a 1 dB reduction in the transmission coupling factor crucial for effective power transfer. The introduction of a 1 mm air gap causes a notable impedance mismatch, resulting in a 4 dB reduction in transmitted power due to the air acting as an additional load. Successive increases in the air gap led to further reductions in the transmitted power and an increase in the power reflected from the air-skin interface. Extending the investigation to analyze transmitted power enhancement using the metasurface within the skin medium, while considering air gap distances of 0.5, 1, and 1.5 mm, demonstrates a decrease in near-field power enhancement inside the skin of 1.2, 4.5, and 7.5 dB, respectively. It is crucial to emphasize that the designated near-field focal point, positioned 2 mm above the metasurface, does not penetrate the skin medium in the presence of an air gap measuring 2 mm or more. Consequently, this non-penetration leads to a lack of observed power enhancement. Under these conditions, the outcomes align with those observed when the metasurface is not integrated.

In scenarios where uniformity is pivotal for deploying the presented metasurface-enhanced radar system in various biomedical applications, several strategies can mitigate the impact of non-uniformity. Firstly, reducing the metasurface array size while preserving resolution enhancement capability addresses non-uniformity concerns by maintaining consistent results in a smaller area. Secondly, incorporating a flexible substrate in the metasurface design allows it to conform to body skin contours, effectively minimizing non-uniformity issues. Thirdly, employing machine learning algorithms, coupled with signal filtering techniques, robustly accounts for skin variability in radar power reception, mitigating interference. Lastly, creating a wideband metasurface enables the radar system to operate across frequencies, offering advantages in penetrating the skin at different depths and interacting distinctively with skin features, potentially providing a more comprehensive perspective.”

Figure 12: (b) Investigation of air gap thickness variations between the metasurface and human skin phantom.

2. When the authors analyse the transmission coefficient inside the radar bandwidth they only focus on the variation of the performance at the central frequency, but it would be important to add also some data in the paper about the variability of this parameter inside the operating band, that can be up to 6 dB. It would also be necessary to clarify if the considered permittivity variation is only related to the real part or also to the conductivity.

Authors' Reply:

We agree that the focus of the first version of the paper was only on the variation of the performance at the central frequency, which is 60 GHz. In the revised version of manuscript, a comprehensive analysis of the transmission coefficient inside the radar bandwidth, 58 to 63 GHz, is presented.

Referring to the revised manuscript in the section “Near-field analysis of the radar system integrated with and without metasurface”, the following was added:

“The preceding investigation pertains to the metasurface unit cell and array, with the analysis conducted at the central frequency of 60 GHz. In light of the designated bandwidth from the

source antenna, the Infineon radar chipset spanning 58 to 63 GHz, it becomes imperative to assess the fluctuations in transmission coefficients across the entire frequency spectrum within the specified range. This comprehensive evaluation is essential for ascertaining the overall bandwidth of the entire system. Fig. 7 illustrates the magnitude and phase responses of the two layers' unit cell for varying lengths of crossed dipole metallic branches. The results are presented across different frequencies within the bandwidth of interest. In Fig. 7c, it is observed that the frequency range from 59 GHz to 61 GHz exhibits a favorable transmission magnitude change, with a maximum reduction of 1 dB. Conversely, at 62 GHz, a higher reduction exceeding 3 dB is observed, indicating that the power improvement technique is less effective at this frequency and further. However, merely examining the transmission coefficient's magnitude is insufficient for determining the bandwidth of the metasurface-enhanced radar design. Consequently, the investigation extends to the transmission coefficient's phase, as depicted in Fig. 7d within the desired bandwidth. Notably, the phase response for frequencies ranging from 59.5 to 61.5 GHz appears nearly parallel, providing a constant shift throughout the bandwidth. Given the satisfactory results obtained from both magnitude and phase analyses of the transmission coefficient in the frequency range of 59.5 to 61.5 GHz, it is anticipated that the near-field-focused radar antenna integrated with the designed metasurface, affords a 2 GHz bandwidth to enhance near-field power within the skin."

Figure 7: (c) magnitude investigation across different frequencies within the desired bandwidth; (d) phase investigation across different frequencies within the desired bandwidth.

Additionally, in the analysis section of the measurement results at the end of the manuscript, the following was added:

"The measurement results showing the power transferred into the water medium and picked up by the probe are presented and compared in Fig. 15a. As can be seen, using the designed transmitarray metasurface enhances the power received by the probe at the operational radar frequency range, from 59.7 to 61.7 GHz. Fluctuations in the enhanced transferred power result from both the distribution of radar power and specific design considerations. Looking specifically at the designed frequency of 60 GHz shows an improvement of 11 dB in the measured near-field power, which is in good agreement with the simulation results representing the power density improvement of 11.5 dB in Fig. 10."

3. Are the results of fig. 11 measured or simulated? If they are measured how is the skin medium realised? If they are simulated, is the water medium also including the shell between the radar and the water slab?

Authors' Reply:

It is essential to emphasize that the findings depicted in Fig. 11 originated exclusively from the simulation process, whereas the measured results are exclusively showcased in Figs. 14 and 15.

In order to enhance the applicability and viability of this study within the realms of manufacturing and measurement processes, the human body skin is conceptualized as a phantom. This phantom is designed to incorporate two interleaved cylindrical dielectric slabs: a Pyrex glass beaker ($\epsilon_r=4.7$ and $\sigma=0.5$ S/m at 60 GHz [59]) filled with pure water ($\epsilon_r=11.17$ and $\sigma=36.4$ S/m at 60 GHz [60]). Moreover, a model of the proposed phantom has been considered into the simulation software, facilitating the comparability of simulation results with measurements.

As such, the proposed phantom not only includes the water slab but also incorporates the shell in the simulation process, given its characterization as a "beaker-filled with pure water." To provide greater clarity, we have updated the legends accompanying Fig. 11. Moreover, this information is elaborated upon in the manuscript, notably in the captions of Fig. 8 and Fig. 11, explicitly stating that the results are derived from simulations and emphasizing the importance of the phantom model.

Figure 11: The S-parameter analysis, simulated reflection and coupling magnitude, of the Infineon radar system integrated with and without the transmissive metasurface in the presence of the proposed phantom, a beaker-filled with pure water.

4. In Fig.11 how is the 9.5 and 11 dB difference at 60 GHz justified by a permittivity variation of 0.77 between the skin and equivalent permittivity of the water+shell? The authors compute an equivalent permittivity (real part) of the water+ shell, but what about the equivalent conductivity? Do the difference between the skin medium and the water medium fall in the 10% permittivity tolerance case. Some clarifications should be added and probably a more representative phantom would be necessary to represent realistic applications of the systems.

Authors' Reply:

An important contribution of the paper is introducing a methodology to substantially enhance radar sensitivity for near-field biomedical sensing, particularly in applications like glucose monitoring. Given the metasurface's direct contact with the skin, precise consideration of the skin's permittivity and conductivity values is essential. In bio-sensing applications involving skin, fat, muscle, and bone, significant differences in dielectric features are common, making it challenging to provide a phantom with accurate features. However, in specific applications such as glucose monitoring or skin cancer detection, where the 60 GHz signal quickly attenuates into the body within a few millimeters, variations in skin dielectric properties do not have a substantial impact.

Nevertheless, for a thorough examination, as clarified in the response to Q1, the revised manuscript now includes relevant information within a dedicated section titled "Sensitivity analysis in near-field-focused bio-sensing design." The initial two sub-sections in this segment delve into the analysis of dielectric features concerning both the proposed phantom and actual human skin.

The following sub-section was added to the manuscript:

“Conducting a comparative analysis between the proposed phantom and human skin is imperative to gain a comprehensive understanding of how well the phantom mimics the dielectric properties of human skin. The analysis enables the identification of any discrepancies or limitations in the proposed phantom, allowing for refinement and improvement.

The difference between the power reflected from the human skin slab and the proposed phantom (beaker filled with pure water), is investigated in Fig. 11. As shown, the metasurface in the presence of the skin model provides 9.5 dB enhancement in the reflected power, whereas the enhancement was 11.5 dB when the beaker was filled with pure water. The difference in the material effective permittivity, ϵ_r , as well as the conductivity, σ , between the human skin model and the beaker filled with pure water accounts for this 2 dB difference in $S(\text{TX-RX3})$ values.

As reported in [60], the permittivity and conductivity of pure water at the frequency of 60 GHz, under typical room temperature conditions, are 11.17 and 65.3 S/m, respectively. Consequently, it becomes imperative to consider the impact of the external shell in the calculations. According to the [59], Pyrex glass demonstrates a relative permittivity of 4.7 and a conductivity of 0.5 S/m, at 60 GHz. In such situations, one can compute the effective dielectric properties of composite materials arranged in series across the entire volume facing the metasurface [61].”

As investigated in Fig. 12a, modifying skin permittivity influences the central frequency and the transmission coefficient within the metasurface unitcell analysis. As illustrated in Fig. 12a, at the operational frequency of 60 GHz, a deviation of $\pm 5\%$ in typical human skin permittivity causes a transmission coefficient reduction of less than 1 dB, while a deviation of $\pm 10\%$ results in reductions of 2 dB and 1.2 dB, respectively. Extending the evaluation to the array structure, it can be shown that altering permittivity by $\pm 5\%$ leads to transferred power reductions of 1.4 dB and 0.9 dB. Modifying permittivity by $\pm 10\%$ results in changes leading to power reductions of 2.3 dB and 1.9 dB.

Therefore, the following was added to the manuscript:

“The effective permittivity of the overall medium, a beaker filled with pure water, is determined to be 8.75. Consequently, comparing the permittivity of the proposed phantom, 8.75, to that of a typical human skin model, 7.98, at 60 GHz reveals a difference of approximately 0.77. When comparing the permittivity of the phantom with that of human skin, it is noted that the tolerance is within +10%. As investigated in Fig. 12a, the anticipated outcome is a 2 dB reduction in the power transferred into the phantom compared to the power transferred into human skin. In this scenario, it can be inferred that the difference in focused power has been transformed into dissipated and reflected power, contributing to an enhancement in reflection power when using the phantom as opposed to the human skin slab. On the conductivity front, the calculated conductivity of the phantom, a beaker filled with pure water, at 60 GHz and room temperature stands at 42 S/m. In contrast, the conductivity of human skin at 60 GHz is measured at 36.4 S/m. This implies that the phantom medium serves as a marginally better conductor, resulting in increased power reflection directed towards the radar.

In conclusion, the examination of both permittivity and conductivity indicates that utilizing the proposed phantom (a beaker filled with pure water) results in less transferred power into the medium compared to the human skin medium, leading to an enhancement in power reflection. This is evident in Fig. 11, where the power reflection from the phantom (11.5 dB) is 2 dB higher than the power reflection from the skin medium (9.5 dB).”

Figure 12: (a) Analyzing human skin permittivity variations and their impact on the transmission coefficient of metasurface unit cells.

5. It seems that the discussion about the lacking of uniformity of the metasurface included in the answer to the reviewer did not result in any modification in the text. This aspect is fundamental since the authors state among other applications the possibility to utilize this technology for example for cancer detection. In this case a different illumination of the target(s) may lead to inaccurate results. Please discuss this in the paper.

Authors' Reply:

The discussion regarding the non-uniformity of the metasurface with human skin is now articulated clearly in the manuscript. As indicated in the response to Q1, the revised manuscript includes relevant

information within a dedicated section titled “Sensitivity analysis in near-field-focused bio-sensing design”. The third sub-section presents a comprehensive analysis of the air gap thickness between the metasurface and human skin. In addition, a graph (Fig. 12b) depicting the dependency of the transmission coefficient on the air gap thickness has been incorporated in the revised manuscript. Additionally, we explicitly specify the maximum tolerated air gap thickness—a critical factor for facilitating power focus within the body and achieving power enhancement. The added sub-section in the manuscript delves into these aspects, offering valuable insights into the impact of air gap variations on our findings.

6. I also confirm that the technical issues from Reviewer 3 have been answered by the authors. For what concerns the first comment about the possible applications of this technology, I agree with Reviewer 3. I appreciate that in the revised version the introduction is improved by providing more context, it would be important to better indicate the advantages but also the limitations of this solution in the cited applications. Also, it would be necessary to relate the choices made during the design process to the issues related to the specific application.

Authors’ Reply:

Thank you for confirming that the technical concerns raised by Reviewer 3 have been adequately addressed. We appreciate your feedback on the revised introduction and your suggestion to further emphasize both the advantages and limitations of our proposed solution in the cited applications.

In response to your valuable input, we enhanced the revised manuscript by providing a more comprehensive discussion on the advantages and limitations of our technology in the specified applications.

The introduced methodology, metasurface-enhanced radar near-field sensing, exhibits versatile applicability across various biomedical sensing applications, including glucose monitoring, skin cancer detection, and on-body radar cardiorespiratory monitoring. Non-invasive monitoring of blood glucose levels with both high sensitivity and resolution has been a significant challenge, as discussed in the existing literature. This revised paper specifically underscores the potential for realizing high-sensitivity mm-wave radar for real-time disease monitoring, with a focus on glucose sensing for diabetic patients as an illustrative example.

The comprehensive vision of this work is to enable glucose monitoring capabilities in wearable devices, as depicted in Fig. 1, such as smartwatches. This emphasis underscores the specific application context for which the metasurface technology is meticulously tailored. The deliberate selection of a 60-GHz center frequency FMCW radar (Infineon, BGT60TR13C) is based on our group's successful prior utilization in near-field probing of human tissue, particularly in the context of glucose level detection [2], [20], [58].

As provided in the manuscript, Fig. 15b offers a detailed analysis of the power received by the radar RX antenna from the proposed phantom (a beaker filled with pure water) in the presence or absence of the designed transmissive metasurface. The varying concentration levels of sugar in the analysis showcase the specific application considerations guiding our design choices. This focused approach aligns the technology with the intended biomedical applications, providing clarity on the relevance and impact of our design decisions.

Comments from Reviewer 2:

The authors have diligently incorporated the feedback from the review process, demonstrating a thorough and thoughtful response to the comments. Consequently, the paper is now considered suitable for acceptance in its current format.

Dear Reviewer,

Thank you for your positive feedback and acknowledgment of the authors' efforts in addressing the review comments. We are pleased to hear that the revisions have been found satisfactory and that the paper is now considered suitable for acceptance in its current format. We appreciate the valuable insights provided during the review process, which have undoubtedly contributed to the enhancement of the paper.

REVIEWERS' COMMENTS:

Reviewer #1 (Remarks to the Author):

The authors addressed all the reviewer's comments. In the current form, the paper is now considered suitable for publication.